# Efficient and safe therapeutic use of paired Cas9-nickases for primary hyperoxaluria type 1

Laura Torella [1], Julia Klermund[2,3,12], Martin Bilbao-Arribas [1,4,12], Ibon Tamayo[4,5], Geoffroy Andrieux[6,7], Kay O Chmielewski[2,3], Africa Vales [1], Cristina Olagüe [1], Daniel Moreno-Luqui[1], Ivan Raimondi[1], Amaya Abad [1], Julen Torrens-Baile[1], Eduardo Salido[8], Maite Huarte [1], Mikel Hernaez[4,5], Melanie Boerries [6,7,9,10], Toni Cathomen [2,3,7,13 ✉], Nerea Zabaleta [11,13 ✉] & Gloria Gonzalez-Aseguinolaza [1,13 ✉]

## Abstract

The therapeutic use of adeno-associated viral vector (AAV)-mediated gene disruption using CRISPR-Cas9 is limited by potential off-target modifications and the risk of uncontrolled integration of vector genomes into CRISPR-mediated double-strand breaks. To address these concerns, we explored the use of AAV-delivered paired *Staphylococcus aureus* nickases (D10ASaCas9) to target the *Hao1* gene for the treatment of primary hyperoxaluria type 1 (PH1). Our study demonstrated effective *Hao1* gene disruption, a significant decrease in glycolate oxidase expression, and a therapeutic effect in PH1 mice. The assessment of undesired genetic modifications through CIRCLE-seq and CAST-Seq analyses revealed neither off-target activity nor chromosomal translocations. Importantly, the use of paired-D10ASaCas9 resulted in a significant reduction in AAV integration at the target site compared to SaCas9 nuclease. In addition, our study highlights the limitations of current analytical tools in characterizing modifications introduced by paired D10ASaCas9, necessitating the development of a custom pipeline for more accurate characterization. These results describe a positive advance towards a safe and effective potential long-term treatment for PH1 patients.

**Keywords** CRISPR-Cas9 Nickase; Metabolic Liver Disease; In Vivo; DSB Repair; AAV Integration
**Subject Categories** Genetics, Gene Therapy & Genetic Disease

## Introduction

Primary hyperoxaluria type 1 (PH1) is a rare autosomal recessive metabolic disease caused by mutations in the *AGXT* gene that encodes the liver-specific peroxisomal alanine-glyoxylate aminotransferase (AGT) that catalyses the transamination of glyoxylate to glycine (Danpure and Jennings, 1986; Cochat and Rumsby, 2013). As a result, glyoxylate accumulates in the cytoplasm and is converted to oxalate by the cytosolic lactate dehydrogenase (LDH) enzyme. Oxalate is a metabolic end product in mammals that needs to be excreted by the renal system. In PH1 patients, urinary oxalate excretion is commonly >1 mmol/1.73 $m^2$ per day, while normal oxalate excretion is <0.45 mmol/1.73 $m^2$ per day. Excess oxalate tends to form insoluble calcium oxalate complexes (CaOx) that crystallize forming stones in the renal tubules, thereby causing kidney damage. This damage leads to manifestations such as urolithiasis, nephrocalcinosis, and eventually progressing to end-stage renal disease (ESRD) (Martin-Higueras et al, 2017). Moreover, with the loss of urinary excretion, the excess of oxalate progressively leads to supersaturation of plasma oxalate and life-threatening systemic oxalosis involving many organs, including bones, heart, retina, and central nervous system (Soliman et al, 2017).

PH1 can be diagnosed at any age, but most patients experience their first symptoms in childhood. Early diagnosis and treatment with conservative therapy may preserve renal function by decreasing oxalate production and crystallization (Cochat and Rumsby, 2013; Martin-Higueras et al, 2016). However, combined liver and kidney transplantations represent the only cure when kidneys are irreversibly compromised (Cochat et al, 2010; Devresse et al, 2020). The combined transplantation presents several limitations, including limited organ availability, the requirement

[1]DNA & RNA Medicine Division, Center for Applied Medical Research (CIMA), University of Navarra, 31008 Pamplona, Spain. [2]Institute for Transfusion Medicine and Gene Therapy, Medical Center - University of Freiburg, 79106 Freiburg, Germany. [3]Center for Chronic Immunodeficiency (CCI), Medical Center - University of Freiburg, 79106 Freiburg, Germany. [4]IdiSNA, Navarra Institute for Health Research, 31008 Pamplona, Spain. [5]Bioinformatics Core, Center for Applied Medical Research (CIMA), University of Navarra, 31008 Pamplona, Spain. [6]Institute of Medical Bioinformatics and Systems Medicine, Medical Center - University of Freiburg, 79110 Freiburg, Germany. [7]Faculty of Medicine, University of Freiburg, 79106 Freiburg, Germany. [8]Hospital Universitario de Canarias, Universidad La Laguna, CIBERER, 38320 Tenerife, Spain. [9]German Cancer Consortium (DKTK), Partner Site Freiburg, 79106 Freiburg, Germany. [10]German Cancer Research Center (DKFZ), 69120 Heidelberg, Germany. [11]Grousbeck Gene Therapy Center, Schepens Eye Research Institute, Mass Eye and Ear, Harvard Medical School, 02114 Boston, MA, USA. [12]These authors contributed equally as second authors: Julia Klermund, Martin Bilbao-Arribas. [13]These authors contributed equally: Toni Cathomen, Nerea Zabaleta, Gloria Gonzalez-Aseguinolaza.
✉E-mail: toni.cathomen@uniklinik-freiburg.de; Nerea_ZabaletaLasarte@meei.harvard.edu; ggasegui@unav.es

for life-long immunosuppression, and high morbidity and mortality rates.

Recently, the inhibition of the glycolate oxidase (GO) enzyme, encoded by the *HAO1* gene and responsible for the synthesis of glyoxylate, has been proven to be an efficacious and safe therapy for severe PH1 patients (Martin-Higueras et al, 2016). In 2017, an RNAi targeting the *HAO1* mRNA demonstrated efficient down-regulation of GO protein levels, which resulted in a dose-dependent reduction of oxalate accumulation, a non-pathological increase of glycolate excretion, and therapeutic efficacy across multiple preclinical models of PH1 (Liebow et al, 2017). In late 2020, the EMA and the FDA approved the RNAi drug, Oxlumo™ (lumasiran) to treat severe PH1 patients. The main limitation of RNAi-based therapies is the requirement for recurring and lifelong administrations, which is associated with potential issues such as poor adherence to the therapy. To address this limitation, CRISPR-Cas9 genome editing can be utilized, which has demonstrated high effectiveness in generating targeted double-strand breaks (DSB) resulting in indel (insertions and deletions) formation and sustained protein expression reduction (Ran et al, 2013b). Recently, this strategy was clinically tested in patients with transthyretin amyloidosis (ATTR) using lipid nanoparticles (LNP) containing a CRISPR-Cas9 system targeting the aberrant transthyretin (TTR) in the liver. Interim results from the first-in-human trial showed a dose-dependent reduction of TTR and significant improvement in symptoms (Gillmore et al, 2021).

Previously, we developed a similar strategy to reduce GO expression in PH1 animals by delivering *Staphylococcus aureus* Cas9 nuclease (SaCas9) using adeno-associated viral (AAV) vectors (AAV-SaCas9-*Hao1*). After the administration of AAV-SaCas9-*Hao1* with a single guide, we showed *Hao1*-specific editing in a high percentage of hepatocytes, greatly diminishing GO expression, and resulting in the reduction of urine oxalate concentration and prevention of kidney damage and nephrocalcinosis, with no signs of toxicity (Zabaleta et al, 2018b). Interestingly, the co-administration of two AAV vectors carrying two different guides targeting the same exon resulted in a perfect deletion of the genomic sequence between the two DSBs and efficient depletion of GO protein (López-Manzaneda et al, 2020). Although no off-target effects were observed for these gRNAs, the approach used to identify them relied on in silico predictions, followed by targeted NGS analysis. This method may not guarantee the utmost accuracy. Moreover, findings related to off-target effects in mice should not be directly applied to humans. Consequently, the potential risk of off-target edits continues to be a concern.

A primary goal for clinical gene editing applications is to minimize off-target effects, and the nickase variant of Cas9 (nCas9) is a promising tool for this purpose (Ran et al, 2013a; Mali et al, 2013b; Kim et al, 2015; Friedland et al, 2015). The cleavage activity of the Cas9 enzyme is mediated by the activities of the HNH and RuvC catalytic domains. While the HNH introduces nick or single-strand break (SSB) in the gRNA complementary strand, cleavage of the noncomplementary strand is mediated by the RuvC domain (Bothmer et al, 2017; Jinek et al, 2012). nCas9 have been developed by the introduction of a single amino acid substitution in either of the two domains: D10A in the RuvC domain and N580A or H557A in the HNH domain of SaCas9 (Friedland et al, 2015; Kleinstiver et al, 2015; Nishimasu et al, 2015). Nicks are sensed by high-fidelity

single-strand break repair pathways typically repaired without introducing sequence errors (Dianov and Hübscher, 2013; Vriend et al, 2016). However, simultaneous nicking of both DNA strands using paired nickases that are appropriately spaced and oriented leads to site-specific staggered DSB formation (Ran et al, 2013a; Mali et al, 2013a). It has been shown that paired nCas9 introduce indels in the target sequence at the same frequencies as Cas9 nuclease (wtCas9), but the probability of off-target modifications is reduced due to the low likelihood of the two nickases binding adjacent off-target sites (Ran et al, 2013a; Gopalappa et al, 2018).

The efficiency of paired nCas9 for precise mammalian genome editing has been characterized in vitro (Trevino and Zhang, 2014; Cho et al, 2014). It has also been employed in mouse zygotes for gene functionality research (Aísa-Marín et al, 2020; Shen et al, 2014) and the generation of animal models (Aísa-Marín et al, 2020; Domènech et al, 2020). In a recent study, hereditary tyrosinemia was successfully corrected in rats using a single nCas9 together with a homology template carried by two independent adenoviruses. However, the initial correction frequency was very low (0.1%) and was amplified thanks to the selective advantage of corrected hepatocytes in this model (Shao et al, 2018). Despite extensive research, the application of paired nCas9 for therapeutic gene disruption in vivo remains unexplored.

Here, we evaluated the therapeutic potential of paired nSaCas9, specifically D10ASaCas9, for mediating *Hao1* disruption in PH1 mice. We performed an in-depth characterization of the genomic modifications at the on-target site and determined the frequency of AAV integration when using single or paired wtSaCas9 versus D10ASaCas9. Our results demonstrate that paired D10ASaCas9 efficiently edited the *Hao1* locus, resulting in reduced GO protein expression, and decreased oxalate accumulation in PH1 mice. Interestingly, we found that the use of paired D10ASaCas9 resulted in heterogeneous on-target modifications, likely repaired by the microhomology-mediated end joining (MMEJ) pathway. As determined by a custom analysis pipeline, the frequency of AAV integration events was significantly reduced while preserving the on-target editing efficacy compared to paired wtSaCas9. Furthermore, we did not detect any off-target modifications or DSB-mediated chromosomal translocations using CIRCLE-seq (Tsai et al, 2017) and CAST-Seq (Turchiano et al, 2021).

# Results

## Paired D10ASaCas9 targeting *Hao1* results in efficient gene disruption in vivo

Here we evaluated the efficacy of the two gRNAs described by Zabaleta et al (2018b) combined with D10ASaCas9 (Fig. 1A). In a previous study (López-Manzaneda et al, 2020), we examined the effectiveness and precision of wtSaCas9 in combination with these gRNAs delivered by two AAV8 at a total dose of $10^{13}$ vg/kg in 12- to 14-week-old PH1 mice that were sacrificed 21 days later. This resulted in indels in 55% of the alleles and reduced GO protein to undetectable levels in the liver. After demonstrating comparable efficacy of paired wtSaCas9 and paired D10ASaCas9 strategies in vitro (Appendix Fig. S1), we evaluated their efficacy in PH1 mice using two AAV8 vectors (Fig. 1B). D10ASaCas9 was expressed under the transcriptional control of the liver-specific TBG and the

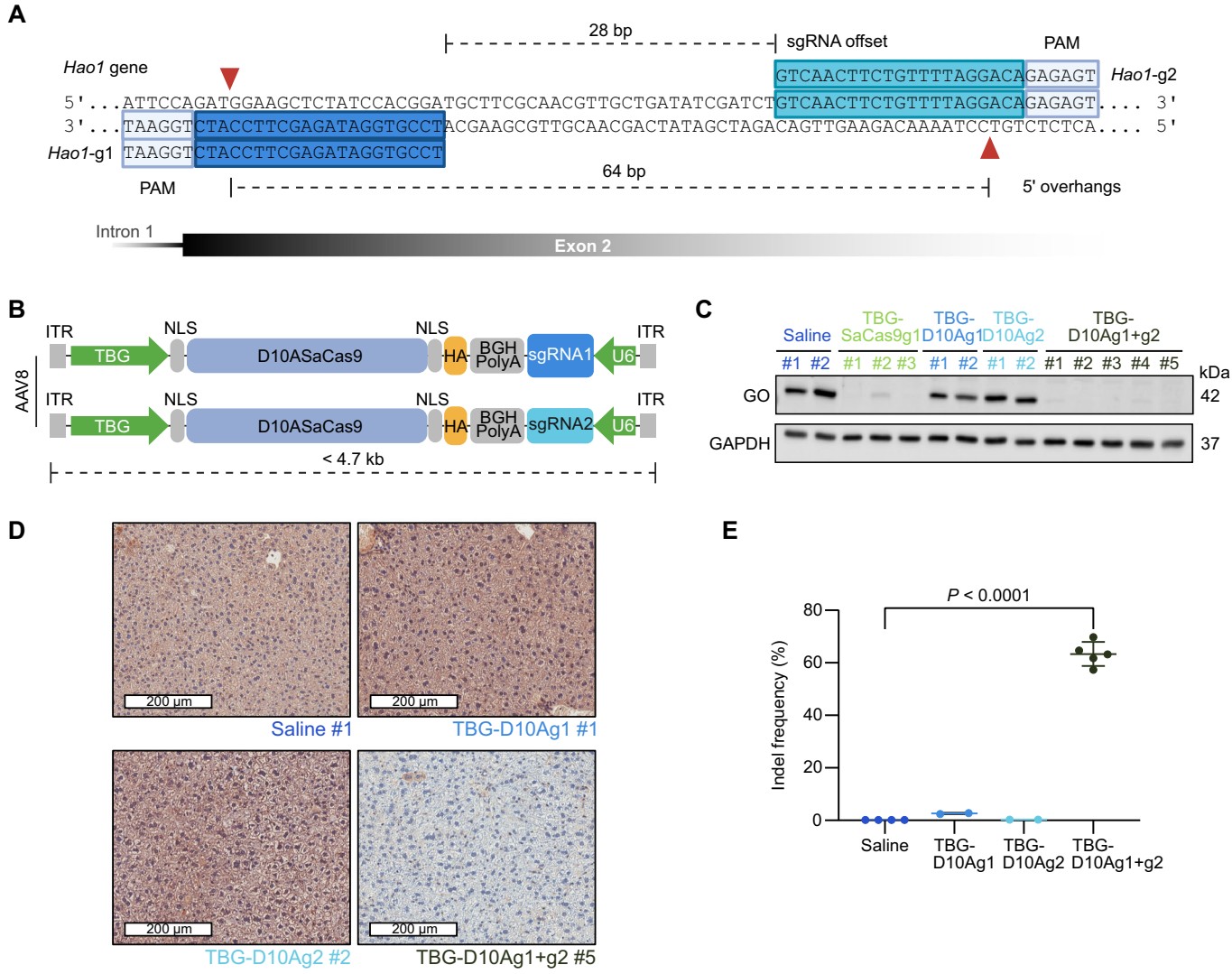

**Figure 1. Efficacy of paired AAV-D10ASaCas9 in PH1 mice.**

(A) Paired D10ASaCas9 strategy design. Red arrows indicate the cut sites of D10ASaCas9. (B) AAV vector design. Inverted terminal repeats (ITR); thyroxin-globulin binding promoter (TBG); nuclear localization sequences (NLS); D10A nickase version of *S. aureus* Cas9 (D10ASaCas9); Bovine growth hormone polyadenylation sequence (BGH pA); hemagglutinin tag (HA); single guide RNA (sgRNA); U6 promoter (U6). (C) WB analysis of GO protein levels. GAPDH was used as a loading control. (D) Representative IHC images of liver sections stained for GO. Scale bar: 200 μm. (E) Frequency of indels assayed by amplicon sequencing. Data information: In (E), data are presented as mean ± SD (Unpaired two-tailed t-test). In (A–E), n = 4–5 mice per group. Source data are available online for this figure.

gRNAs under the control of the U6 promoter. PH1 mice were divided into five different groups and received an intravenous injection of: (I) saline solution (saline), (II) $5 \times 10^{13}$ vg/kg AAV-wtSaCas9-*Hao1*-g1 (TBG-SaCas9g1) (Zabaleta et al, 2018b), (III) $5 \times 10^{13}$ vg/kg AAV-TBG-D10ASaCas9-*Hao1*-g1 (TBG-D10Ag1), (IV) $5 \times 10^{13}$ vg/kg AAV-TBG-D10ASaCas9-*Hao1*-g2 (TBG-D10Ag2), or (V) $5 \times 10^{13}$ vg/kg total of AAV-TBG-D10ASaCas9-*Hao1*-g1 and AAV-TBG-D10ASaCas9-*Hao1*-g2 (TBG-D10Ag1 + g2). Four weeks after vector injection mice were sacrificed and the expression of GO protein in the liver was analysed by western blot (WB) and immunohistochemistry (IHC). As demonstrated by WB, GO expression was reduced to undetectable levels in all the animals treated with TBG-D10Ag1 + g2, and in most of the mice that received TBG-SaCas9g1, while GO levels remained unaltered in the remaining groups (Fig. 1C and Appendix Fig. S2A). The WB results were corroborated by IHC, by which none or very scarce expression was detected in TBG-D10Ag1 + g2-treated mice while a homogenous expression of GO was detected in the animals receiving saline solution, TBG-D10Ag1 or TBG-D10Ag2 (Fig. 1D and Appendix Fig. S2B). Sequencing analysis of the on-target genomic region revealed that between 57.4% and 69.8% of *Hao1* alleles were edited in the animals that received TBG-D10Ag1 + g2 (Fig. 1E). This frequency is comparable to the one achieved by wtSaCas9 with *Hao1*-g1 or *Hao1*-g2 (Zabaleta et al, 2018b) or a combination of the two (López-Manzaneda et al, 2020), but using a 10- and 5-fold higher dose of vector respectively (Zabaleta et al, 2018b;

López-Manzaneda et al, 2020). A very low frequency of gene editing was detected in TBG-D10Ag1-treated mice (2.5–2.8%) and none in TBG-D10Ag2 and saline groups.

## Paired nicks are primarily repaired by MMEJ and lead to a lower AAV integration rate in comparison to nuclease Cas9-mediated DSBs

In our previous publication (López-Manzaneda et al, 2020), we used Cas-Analyzer and a minimum frequency $n = 100$ for each specific event (Park et al, 2017), however, using this tool with the same specifications we were unable to determine the modifications introduced by TBG-D10Ag1 + g2, which are characterized by high heterogeneity and a very low frequency of occurrence for each event (Fig. 2A). Therefore, we developed a custom preprocessing pipeline using the CrispRVariants R package (Lindsay et al, 2016) to uniformly analyse the amplicon sequencing data from our current study as well as reanalysed our previous data (López-Manzaneda et al, 2020). We further optimized this pipeline to address the inefficiency of CrispRVariants, and similar tools, in processing the long insertions resulting from AAV genome integration. The analysis revealed that the use of TBG-D10Ag1 + g2 resulted in both insertions and deletions of variable sizes (Fig. 2A,C,D and Appendix Fig. S3A,B). The most frequent indels are larger than 10 bp and occur between the two nicking sites. Predominantly, deletions range from 20 to 30 bp in length and insertions range between 5 and 25 bp and match the sequences between the two nicks (Fig. 2A,D and Appendix Fig. S3A,B). In addition, we detected multiple insertions or deletions as well as combinations of both (Fig. 2C and Appendix Fig. S3B). Furthermore, upon analysing the NGS data obtained from animals treated with TBG-SaCas9g1 + g2 using the custom pipeline, we not only detected the expected 64 bp deletion (31–38.5% of total indels) but also observed shorter deletions, small insertions, and a variety of longer insertions (Fig. 2B–D and Appendix Fig. S4A,B). Moreover, upon analysis of the raw data published in Data ref: Zabaleta et al (2018a), which utilized single guide SaCas9, it was confirmed that the use of TBG-SaCas9g1 primarily resulted in deletions and insertions smaller than 25 bp, with an indel frequency of 51.6%, while TBG-SaCas9g2 showed a high tendency to induce deletions smaller than 25 bp, with an indel frequency of 53.78% (Fig. 2C,D and Appendix Fig. 5A,B).

Interestingly, we found that animals treated with TBG-D10Ag1 + g2 exhibited an average AAV vector sequence integration rate of 1.7% at the on-target site, which accounted for 3% of total indels (Fig. 2E). In contrast, animals treated with TBG-SaCas9g1 + g2 had a significantly higher AAV integration rate of 9.8% of total reads, corresponding to 16.4% of total indels (Fig. 2E). These differences are particularly surprising taking into consideration that TBG-D10Ag1 + g2-treated mice received five times more AAV vector genomes than those treated with TBG-SaCas9g1 + g2. When using TBG-SaCas9g1 or TBG-SaCas9g2, the frequency of AAV integration was similar to that detected in animals treated with the combination of the two gRNAs. We observed a higher frequency of AAV integration in animals treated with TBG-SaCas9g1 (12.6%) than in animals receiving TBG-SaCas9g2 (7.7%). This result is consistent with the fact that *Hao1*-g1 is more prone to insertions (Fig. 2E; Appendix Figs. S5A,B and S6A). No AAV sequences were detected at the on-target site in animals receiving

TBG-D10Ag1 or TBG-D10Ag2 (Fig. 2E). The analysis of integrated AAV sequences showed that they mainly corresponded to truncated inverted terminal repeats (ITRs) (Fig. 2F and Appendix Fig. S6B), with a preferred breakpoint between the B′–B and A′–A arms common to the 3' and 5' ITRs (Fig. 2G and Appendix Fig. S6C).

Finally, we analysed the NGS data using CRISPResso (Pinello et al, 2016) followed by mhscanR (Owens et al, 2019) to determine the frequency of MMEJ and to better characterize the nature of the on-target modifications introduced by the different editing systems. This analysis revealed that animals treated with TBG-D10Ag1 + g2 had a significantly higher frequency of MMEJ events of more than 2 base pairs (bp) compared to the animals treated with TBG-SaCas9 (Fig. 2H,I). The DSBs introduced by paired or single SaCas9 were mainly repaired in the absence of microhomology (MH) or the presence of an MH of 1 bp (Fig. 2H,I). Interestingly, all animals treated with paired D10ASaCas9 showed two MMEJ events with a higher frequency than any other indel: a deletion of 23 bp associated with a 4 bp MH and a deletion of 10 bp associated with a 5 bp MH (Fig. 2J).

## The minimal therapeutic dose is reduced when combining D10ASaCas9 and the two gRNAs in a single vector

Next, to facilitate the clinical translation of this therapeutic approach, we developed an all-in-one AAV vector carrying all the elements required for targeted *Hao1* cleavage, i.e. D10ASaCas9 with the two gRNAs and the corresponding regulatory sequences. Due to the limited packaging capacity of the AAV vectors, the TBG promoter and the bovine growth hormone (BGH) polyadenylation signal (pA) were substituted by the human alpha-1-antitrypsin (AAT) promoter and a short synthetic pA signal, respectively (Fig. 3A upper). For comparative purposes, we generated AAV vectors carrying the same elements but only one of the guides that were co-administered (Fig. 3A lower).

Three different doses of (I) $5 \times 10^{13}$ vg/kg (high dose or HD), (II) $1.5 \times 10^{13}$ vg/kg (medium dose or MD), and (III) $5 \times 10^{12}$ vg/kg (low dose or LD) of the all-in-one vector (AAT-D10Ag1g2) were tested in PH1 mice and compared to PH1 mice receiving the combination of the two independent vectors (AAT-D10Ag1 + g2) at a total dose of $5 \times 10^{13}$ vg/kg (HD) and $5 \times 10^{12}$ vg/kg (LD) or only one vector (AAT-D10Ag1; AAT-D10Ag2). A group of PH1 animals that received saline solution was included as control (saline). The analysis of GO expression by IHC showed a few hepatocytes expressing GO in the animals treated with either of the three different doses of AAT-D10Ag1g2 and with the high dose of the two independent vectors AAT-D10Ag1 + g2, while the expression was more abundant in the animals treated with the low dose of AAT-D10Ag1 + g2 (Fig. 3B and Appendix Fig. S7A). WB revealed undetectable protein in the mice that received the two highest doses of the AAT-D10Ag1g2, and a very faint protein band was detected in some of the animals treated with the LD of this vector (Fig. 3C and Appendix Fig. S7B). When co-administering the vectors carrying one of the gRNAs, the animals treated with the HD had undetectable levels of GO protein. Conversely, GO protein was clearly detected in the animals receiving the LD (Fig. 3C and Appendix Fig. S7A,B). The animals treated with only one gRNA showed the same GO protein level as the saline controls.

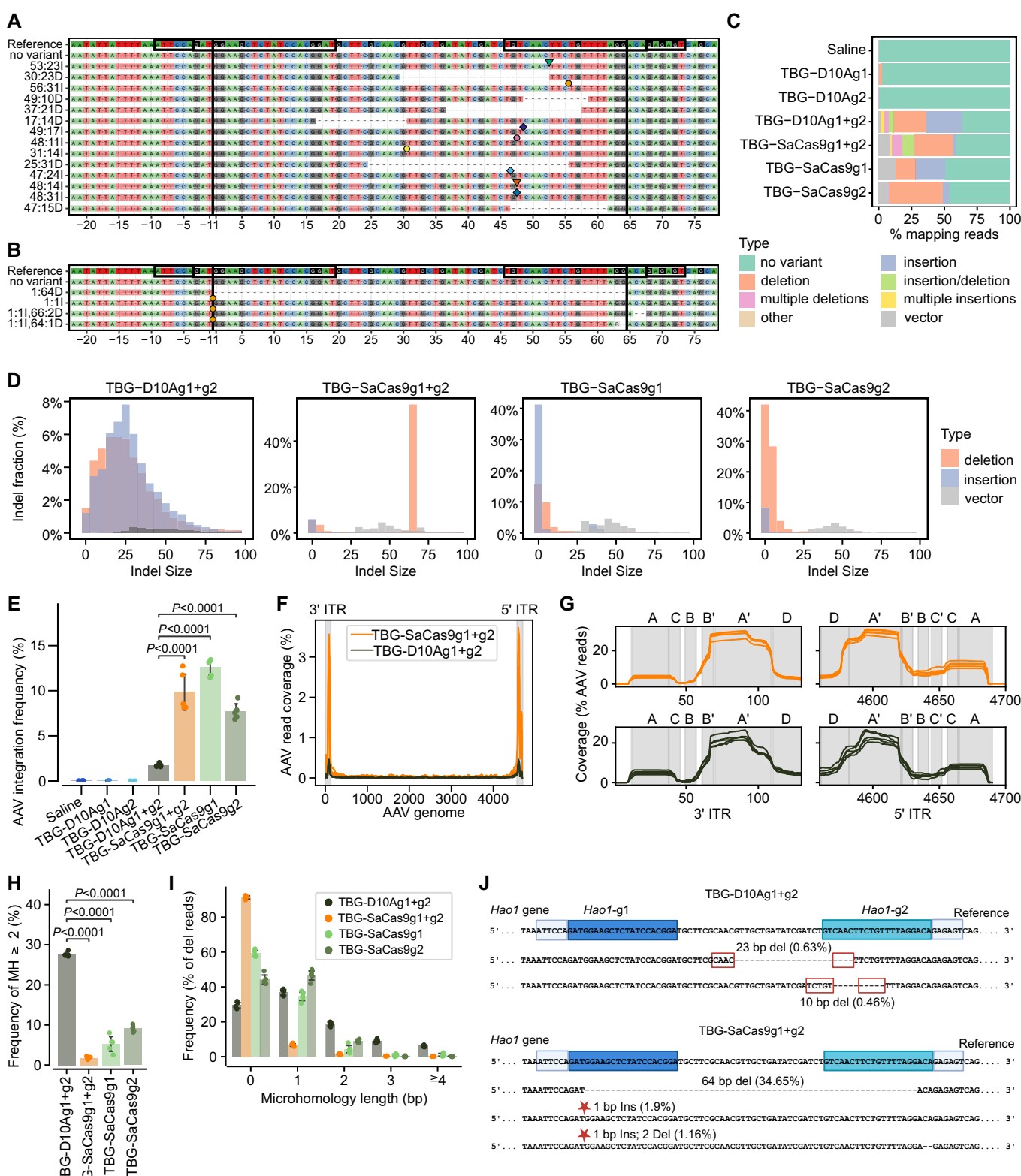

Sequencing analysis of the on-target genomic region (of randomly selected animals) detected very similar frequencies of allele modification (38.9% to 58.7%) in the animals receiving the all-in-one system independently of the dose, indicating that a maximum effect was achieved at the lowest dose (Fig. 3D). In those treated with AAT-D10Ag1 + g2 at HD, the indel frequency was comparable to that achieved with the all-in-one vector. However, at LD there was a trend ($p$-value = 0.09) for a lower editing frequency,

**Figure 2. Characterisation of on-target genomic modifications by TBG-D10ASaCas9 and TBG-SaCas9 in PH1 animals.**

(A, B) Summary of the most frequent variant types and locations in PH1 mice injected with TBG-D10Ag1 + g2 (A) and TBG-SaCas9g1 + g2 (B). Black blocks represent the target site and PAM sequence, while black lines indicate the expected cut site. Indel size and location are given on the left, with deletions as dashed lines and insertions as coloured symbols. Representative indel variants are plotted by frequency. (C) Bar chart of allele variant frequency (mean). (D) Indel size distribution: orange for deletions, blue for insertions, and grey for insertions of AAV sequences (vector). (E) AAV integration frequencies on-target. (F) Coverage of integrated sequences across the AAV genome sequence. (G) Coverage of 5′ and 3′ ITR integrated regions with a preferred breakpoint between the B′–B and A′–A-arms. (H) Frequency of microhomologies (MHs) larger than 2 bp among the deletions. (I) Frequency distribution of deletions (del) according to the size of MHs. (J) Representation of the most frequent modifications. The reference sequence is shown in the first line and gRNA1 and gRNA2 are highlighted. PAM sequences are underlined in light blue boxes, deletions represented as dashed lines and insertions as stars. MH sequences are in bold and surrounded by a red box. Data information: In (D–I), data are presented as mean ± SD. In (E, H), one-way ANOVA followed by Tukey's post-hoc test. In (A–J), n = 4–5 mice per group.

indicating a dose-dependent effect (Fig. 3D). In addition, as observed with the TBG vectors, very low levels of on-target modifications (average 1.5%) were detected in the animals treated with either AAT-D10Ag1 or AAT-D10Ag2 (Fig. 3D). Finally, we found that liver transduction was dose-dependent and equivalent in the groups that received the same dose of either AAT-D10Ag1g2 or AAT-D10Ag1 + g2 (Fig. 3E), indicating that the editing efficacy was improved when using the all-in-one system.

To evaluate the therapeutic potential of AAT-D10Ag1g2, PH1 mice were treated with high, medium, and low doses of the vector, and one month later the animals were challenged by the administration of 0.5% ethylene glycol (EG) in drinking water for 10 consecutive days (EG is a precursor of glyoxylate that increases oxalate production) (Salido et al, 2011; Dutta et al, 2016) (Fig. 3F). Oxalate concentration was determined in 24-h urine samples harvested at baseline (prior to the EG challenge) and on days 3 and 7 after EG administration (Fig. 3F). Saline-treated PH1 and WT (C57BL/6J) mice were included as controls. As shown in Fig. 3G, oxalate levels in urine prior to the EG challenge were significantly lower in the three groups of animals that received the all-in-one system than in saline-treated PH1 mice. These differences increased after the EG challenge (Fig. 3G). Moreover, after the challenge oxalate levels were similar in all-in-one treated animals, regardless of the dose, and comparable to the one measured in healthy WT mice (Fig. 3G). These results are consistent with the levels of GO protein shown in Fig. 3B,C. Furthermore, the animals that received the all-in-one system had a higher concentration of glycolate in urine, as expected from the mechanism of action of the treatment (Fig. 3F,H).

## Minimal AAV integration frequency is observed when using the all-in-one system

The analysis of the sequencing data revealed that the all-in-one system resulted in both insertions and deletions of variable sizes, similar to previous findings with the co-administration of the two vectors (Fig. 4A,B and Appendix Fig. S8A,B). The most frequent modifications were deletions larger than 10 bp, which occurred between the two nicking sites (Fig. 4A and Appendix Fig. S8A) and exhibited the same pattern as the dual vector approach shown in Fig. 2D (TBG-D10Ag1 + g2).

The AAV integration frequency at the on-target site was notably low with both the all-in-one vector and the two AAT vectors, ranging from 0.03% to 0.82% (Fig. 4C). This frequency was lower than that observed in animals receiving the TBG-D10Ag1 + g2 vectors, which was 1.7% (Fig. 2E), representing a 2-fold difference. This difference might be attributed to experimental variations that resulted in minor differences in the number of AAV genome copies

(Appendix Fig. S9A) and Cas9 expression (Appendix Fig. S9B) in the livers of mice injected with the same dose ($5 \times 10^{13}$ vg/kg). Interestingly, even though there were no significant differences in the editing efficiency between the three doses, a dose effect was detected in the AAV integration frequency, which was significantly higher in the animals receiving the higher doses. Therefore, the lowest dose of the all-in-one vector resulted in an optimal editing profile, with high indel frequency and low vector integration on-target (Fig. 4C). As previously shown, the integrated AAV sequences corresponded predominantly to truncated ITRs (Fig. 4D). The analysis of MH length and frequency revealed a high frequency of events associated with MMEJ in all the groups (Fig. 4E,F). The animals receiving the lowest dose of either treatment showed a trend to a higher frequency of deletions associated with larger MH domains than animals receiving the higher doses (Fig. 4E,F). Once again, the two most frequent MMEJ modifications were a 23 bp and a 10 bp deletion (Appendix Fig. S8A).

## CIRCLE-seq and CAST-Seq analyses revealed neither off-target activity nor chromosomal translocations

In our previous study, we analysed potential off-target modifications by deep sequencing of predicted off-target regions (using Benchling software) for each gRNA and detected no off-target modifications (Zabaleta et al, 2018b). Because of the limited sensitivity of in silico prediction assays, we performed two additional off-target analyses using CIRCLE-seq (circularization for in vitro reporting of cleavage effects by sequencing) (Tsai et al, 2017) and CAST-Seq (chromosomal aberration analysis by single targeted LM-PCR sequencing) (Turchiano et al, 2021). CIRCLE-seq is a powerful in vitro assay able to identify SaCas9-induced off-target DSBs independently of their genomic context, overcoming the sensitivity limitations of cell-based methods. Despite its high sensitivity, we detected only a few potential off-targets for both guides and at very low frequencies (Fig. EV1). Moreover, most of these sequences had low identity (63–74.1%) with the guide sequences and some of them had mismatches in the protospacer adjacent motif (PAM) sequence (NNGRRT), which dramatically reduces the possibility of happening in vivo. The identified off-target sites for *Hao1*-g1 had mismatches in the PAM proximal regions and/or in the PAM sequence, which excludes them as functional off-targets. For *Hao1*-g2, the off-target site with the highest score located in chromosome 9 (chr.9) contained mismatches mainly in the PAM distal region and had the consensus PAM sequence. Therefore, in order to validate this off-target effect in liver samples, we attempted to amplify this region. However, we faced a challenge due to the highly repetitive nature of the target region, which hampered us from successfully amplifying the identified off-target site.

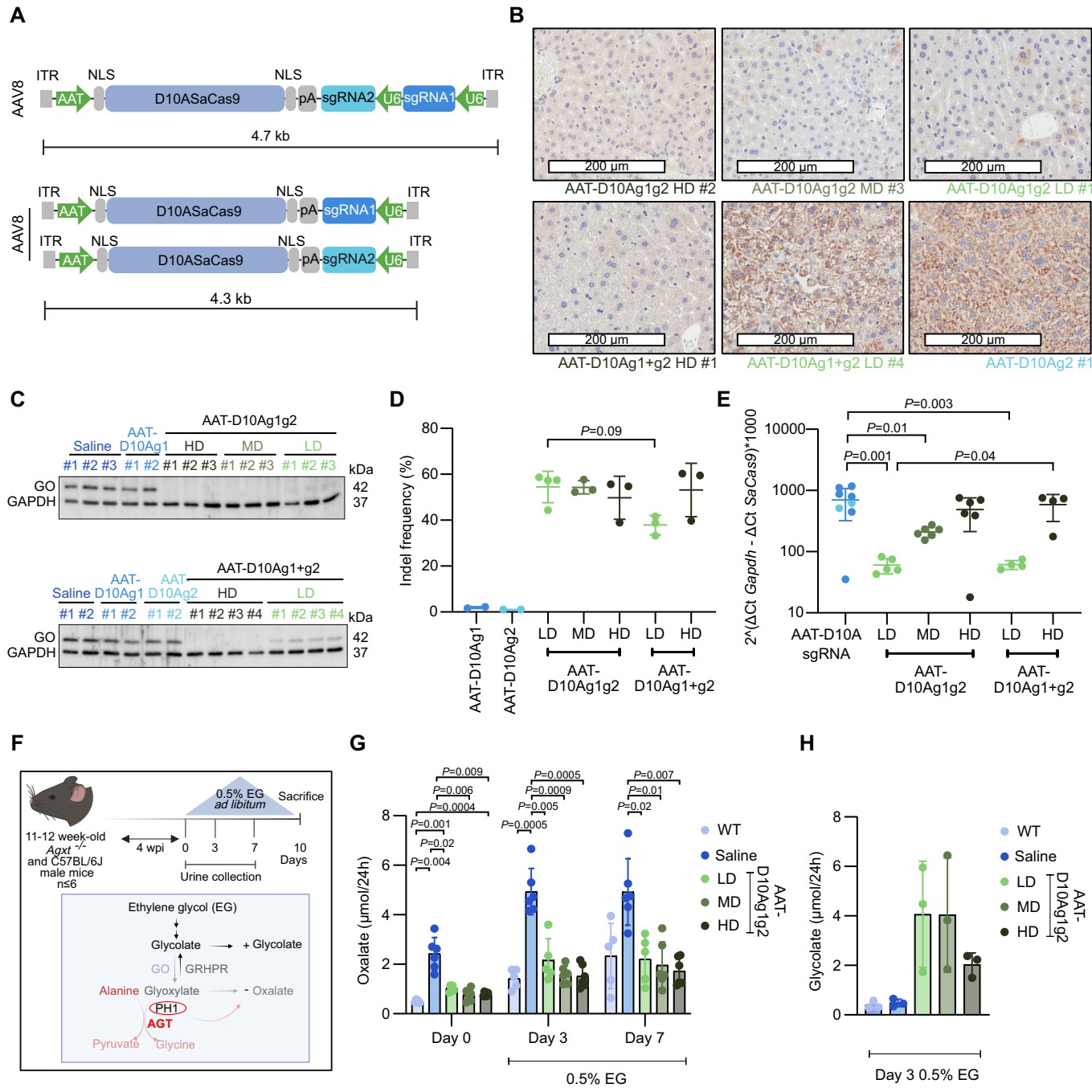

**Figure 3. In vivo therapeutic efficacy of the all-in-one paired nickases system in PH1 animals.**

(A) All-in-one AAV vector (top); and individual AAV vector constructs (bottom). Inverted terminal repeats (ITR); nuclear localization sequences (NLS); human alpha-1-antitrypsin promoter (AAT); D10A nickase version of *S. aureus* Cas9 (D10A*Sa*Cas9); small synthetic polyadenylation sequence (pA); single guide RNA (sgRNA1 and sgRNA2); U6 promoter (U6). (B) Representative IHC images of liver sections stained for GO. Scale bar: 200 μm. (C) WB analysis of GO protein expression in all-in-one treatment (top gel) and in the two-vector system (bottom gel). GAPDH was used as a loading control. (D) Quantification of the percentage of indel frequency. (E) Relative quantification of viral genome copies in the liver of treated animals. (F) Schematic representation of the experimental procedure and description of the oxalate pathway in PH1 mice after GO knockdown. (G) Measurement of oxalate (μmol/24 h) in 24-h urines before challenge (Day 0) and on day 3 and day 7 of EG challenge in C57BL/6J injected with saline and PH1 mice injected with different vectors or saline; 4 weeks-post injection (4 wpi). (H) Measurement of glycolate levels (μmol/24 h) in 24-h urine samples on day 3 of EG challenge. Data information: In (D, E, G, H), data are presented as mean ± SD. In (E, H), one-way ANOVA followed by Tukey's post-hoc test. In (G), two-way ANOVA followed by Tukey's post-hoc test. In (A–H), $n = 2$–6 mice per group. Source data are available online for this figure.

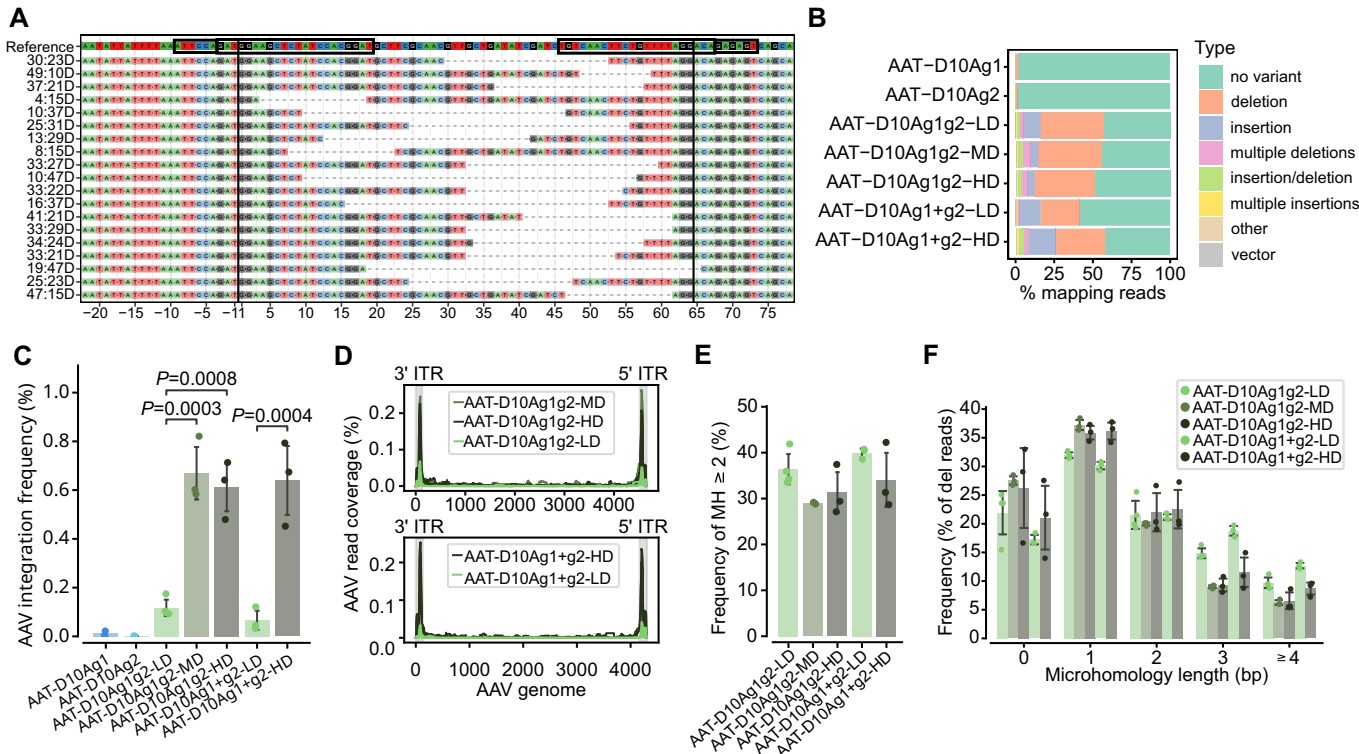

**Figure 4. Characterisation of on-target genomic modifications in PH1 animals treated with the all-in-one system or the two-vector system.**

(A) Summary of the most frequent variant types and locations in PH1 mice injected with AAT-D10Ag1g2 and AAT-D10Ag1 + g2. Black blocks represent the target site and PAM sequence, while black lines indicate the expected cut site. Indel size and location are given on the left, with deletions as dashed lines. (B) Bar chart of allele variant frequency (mean). (C) AAV integration frequencies on-target. (D) Coverage of integration sites across the AAV genome sequence (mean). (E) Frequency of microhomologies (MHs) larger than 2 bp among the deletions. (F) Frequency distribution of deletions (del) according to the size of MHs. Data information: In (C–F), data are presented as mean ± SD. In (C), one-way ANOVA followed by Tukey's post-hoc test. In (E), one-way ANOVA not significant $p = 0.068$. In (A–F) $n = 2$–6 mice per group.

Next, we explored the use of CAST-Seq to detect gross chromosomal aberrations and to identify off-target sites following *Hao1* editing. CAST-Seq uses a bait primer to detect sequences fused to the on-target site as a result of large deletions or inversions at the on-target site, or from translocations due to off-target cleavage in different chromosomes. Decoy primers at the opposite side of the on-target site with respect to the bait primer prevent amplification of sequences that are unedited or harbour small indels (CAST-Seq schematic overview in Appendix Fig. S10A and optimization of bait and decoy primers in Appendix Fig. S10B). Mouse embryonic fibroblasts (MEFs) were nucleofected with SaCas9 ribonucleoprotein (RNP) complexes, using either one guide alone (SaCas9g1 or SaCas9g2) or the combination of the two guides (SaCas9g1 + g2) (Fig. 5A). Genomic DNA was harvested 6 days later, and on-target activity was determined by T7E1 assay (Appendix Fig. S10C) and NGS analysis (Fig. 5B,C and Appendix Fig. S10D). Indel frequencies were 58.4% for SaCas9g1, 64.9% for SaCas9g2, and 99.5% for the combination of SaCas9g1 + g2 (59.4% of alleles with a perfect 64 bp deletion) (Fig. 5B,C). CAST-Seq analysis revealed that both SaCas9/gRNA combinations were highly specific and led mostly to on-target aberrations (Fig. 5D). A translocation with chr.8 was identified in all three experimental setups but classified as an off-target-mediated translocation (OMT)

only in SaCas9g2-treated samples (Dataset EV1). The reads that were fused to the on-target site stemmed from a ∼7 kb region on chr.8 (Fig. EV2A) and not from a particular site. Within this 7 kb region, only in the SaCas9g2-treated samples the read coverage peaked in a region with a moderate match to a putative target sequence for g2. However, the predicted off-target cleavage site for SaCas9g2 had a high number of mismatches and non-matching PAM (Dataset EV1), suggesting a CAST-Seq artefact. Long-read Nanopore sequencing covering a 5 kb region on chr.8 did neither reveal off-target editing in the edited MEFs nor in the in vivo edited hepatocytes (Fig. EV2B), confirming that the observed translocations between *Hao1* locus and chr.8 are most likely a CAST-Seq artefact. In summary, the combination of CAST-Seq, CIRCLE-seq, and Nanopore sequencing confirmed the high specificity of the selected gRNAs for the *Hao1*-targeting approach.

# Discussion

CRISPR-Cas9 gene editing in vivo has emerged as a promising clinical approach, as evidenced by the successful clinical trial for patients with transthyretin amyloidosis (ATTR) (Gillmore et al, 2021), opening new doors for precision medicine. The results of

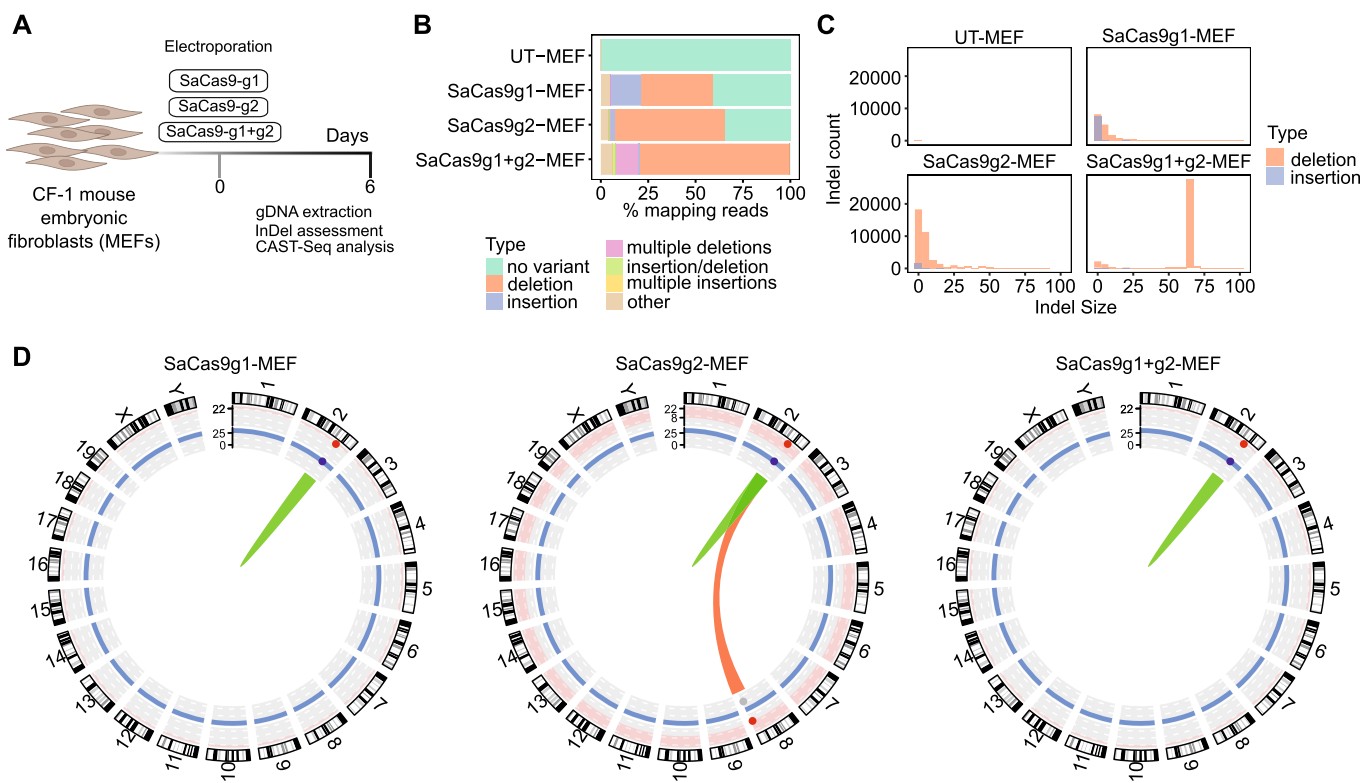

**Figure 5. Detection of CRISPR-mediated chromosomal translocation in nucleofected MEFs.**

(A) Schematic representation of the electroporation performed in CF-1 mouse embryonic fibroblast (MEFs) and downstream molecular analysis performed on harvested gDNA. Three RNP gRNA mixes were nucleofected (SaCas9-g1, SaCas9-g2, and SaCas9g1 + g2, respectively). (B) Bar chart with the frequency of allele variants. (C) Indel size distribution of deletions and insertions. Deletions are shown in red blocks and insertions in blue. (D) Circos-plots summarizing CAST-Seq analyses performed in electroporated MEFs. On-target (ON) rearrangements are displayed in green, and off-target mediated translocations (OMT) are in red. Data information: In (A), $n = 2$ biological replicates for each condition. In (B, C), $n = 1$ biological replicate for each condition. In (D), SaCas9g1 $n = 1$ biological replicate, 2 technical replicates, SaCas9g2 $n = 2$ biological replicates with 2 technical replicates each, SaCas9g1 + 2 $n = 3$ biological replicates with 2 technical replicates for 1 of them. Source data are available online for this figure.

this trial confirmed previous preclinical studies in animal models, in which liver cells were modified by the introduction of necessary genome editing components. This resulted in a significant reduction of transthyretin and substantial clinical benefits. Previously, we used a similar strategy to develop a substrate reduction therapy (SRT) for primary hyperoxaluria type 1 (PH1) (Zabaleta et al, 2018b). In this study, we demonstrated that a sustained reduction of GO protein expression could be achieved in PH1 mice through the administration of an AAV vector carrying both SaCas9 and *Hao1*-specific gRNAs, resulting in the reduction of oxalate production and prevention of nephrocalcinosis.

However, one of the major caveats of using the CRISPR-Cas9 editing system is the potential for off-target effects. In the current study, we substituted the SaCas9 nuclease with its nickase variant (D10ASaCas9) to enhance the safety of our gene disruption approach. D10ASaCas9 was specifically selected for its superior performance compared to the HNH nicking counterpart, as previously demonstrated in vitro (Friedland et al, 2015; Bothmer et al, 2017). We anticipate similar outcomes in terms of indel formation with N580A or H557A since in non-dividing cells, such as quiescent hepatocytes, DSBs will undergo repair via the same mechanism (Bothmer et al, 2017). However, the indel distributions

might differ with N580A resulting predominantly in insertions and D10A indels evenly split between insertions and deletions (Friedland et al, 2015).

D10ASaCas9 was delivered by AAVs in combination with two gRNAs targeting exon 2 of the *Hao1* gene at a distance of 64 bp in PAM-out orientation and targeting opposite strands. With this approach, only simultaneous nicking of both DNA strands would lead to site-specific DSB formation. Variability in the gRNAs distance can profoundly influence indel rates. According to Friedland et al (2015), the 64 bp falls within the optimal range for efficient indel formation, which offset spans from 0 to 170 bp. It has been previously demonstrated that D10ACas9-induced nicks are mostly repaired by the high-fidelity homologous recombination (HR) machinery or by alternative pathways such as MMEJ (Dianov and Hübscher, 2013; Vriend et al, 2016) with indel rates ranging between 0% and ~8% (Friedland et al, 2015). Our study confirmed that the administration of a single AAV vector carrying a single gRNA along with D10ASaCas9 led to minimal or no detectable frequencies of on-target modifications and did not cause any significant alteration in the expression levels of the GO protein. However, when the two vectors were co-administered, we observed an indel frequency of more than 50% and a significant reduction in

protein expression. Furthermore, we optimized the system by including all gene-editing components in a single vector. Using this all-in-one system, we found that with a relatively low vector dose, we were able to reduce GO expression to undetectable levels and, more importantly, prevent oxalate accumulation in PH1 mice. As expected from the mechanism of action of this therapy, GO inhibition resulted in an increase of glycolate, which is easily eliminated via urine. The therapeutic efficacy achieved at the lowest tested dose demonstrated that, with the all-in-one construction, there is no genuine requirement for a dose higher than $5 \times 10^{12}$ vg/kg, same dose determined to be efficacious for the nuclease version (Zabaleta et al, 2018b). Interestingly, the efficacy of indel formation and GO inhibition was quite similar in animals treated with the three different doses of the all-in-one vector. However, there was a trend toward lower efficacy in animals that received the two independent vectors at the low dose. We hypothesize that the superior efficacy of the all-in-one system stems from the simultaneous expression of both guides within the same cell. In contrast, with the two independent vectors, simultaneous co-expression of the guides might occur in fewer hepatocytes or alternatively the amount of each gRNA in the cell might not be equal, and this imbalance could hinder the availability of both gRNAs for simultaneous nicking.

It is known that some genome editing analysis tools are not able to properly process insertions longer than those created by NHEJ repair (Labun et al, 2019), a trait that is critical to properly identify the genome modifications and for the assessment of vector sequence integration. To solve these issues, we tried several tools to characterize the editing outcomes of AAV-mediated delivery of wtSaCas9 or D10ASaCas9: CRISPResso2 (Clement et al, 2019), CrispRVariants (Lindsay et al, 2016), campliCan (Labun et al, 2019), and CRISPECTOR (Amit et al, 2021). However, none of these tools were able to fully capture all the AAV sequence integration events, especially in the case of D10ASaCas9. Therefore, we developed an analysis pipeline based on the CrispRVariants (Lindsay et al, 2016) tool but optimized to retain and quantify all the reads that contain AAV sequences. This strategy was used to analyse both the new and our previously published NGS data obtained with wtSaCas9 (Zabaleta et al, 2018b; López-Manzaneda et al, 2020). We found that the on-target genetic modifications introduced by D10ASaCas9 and the two guides were highly variable, in contrast to the predominant deletion of 64 bp observed when using the wtSaCas9 combined with the same two guides.

The features of the DNA ends generated by the different Cas9 variants are crucial for the repair pathway implicated in the resolution of DSBs, and hence the editing outcomes (Xue and Greene, 2021). The repair of DNA DSBs relies primarily on whether DNA end resection occurs. Canonical NHEJ (c-NHEJ) should be the major pathway used to repair blunt ends or 1–2 bp 5′ staggered ends created by wtCas9 nucleases, mainly resulting in 1 to 4 bp deletions and insertions. However, when staggered ends are created, like those predicted to occur with paired D10ASaCas9, alternative pathways can compete for the repair of DSBs (Ceccaldi et al, 2016). Since homologous recombination (HR) is not always available and the staggered ends are relatively short for single-strand annealing (SSA), MMEJ is predicted to be the main mechanism of DSB repair and indel formation in these situations (Bhargava et al, 2016; Ahrabi et al, 2016). We analysed the extent of MHs around the cut sites and detected a high frequency of MH events in paired D10ASaCas9-treated mice, but very few in

wtSaCas9 mice. Interestingly, although we observed high variability in the sequences obtained after paired Cas9 nickases cleavage, we found a higher frequency of two deletion events associated with 4 and 5 bp MHs. Moreover, as previously demonstrated (Hanscom et al, 2022), MMEJ can result in insertions larger than 5 bp, and the inserted sequence mainly corresponds to the sequence within 50 nt of the DSB. In our case, the most frequent insertions identified in mice treated with paired D10ASaCas9 are indeed larger than 5 bp and match with the region in between the two nicks. These results indicated that the 5' staggered ends were mainly repaired by the MMEJ pathway.

AAV sequence integration into CRISPR-induced DSBs is relatively frequent when using wtCas9, as previously reported by others (Breton et al, 2020; Hanlon et al, 2019; Simpson et al, 2023). The analysis of the AAV integration frequency upon treatment with paired D10ASaCas9 and wtSaCas9 demonstrated interesting differences: while the frequency of indels was very similar, the frequency of AAV integration was significantly higher with wtSaCas9 than with paired D10ASaCas9. Importantly, the use of a lower dose of the all-in-one D10ASaCas9 vector resulted in high editing efficacy with nearly undetectable AAV insertions. This suggests that the type of Cas9 used has an impact on the frequency of genome integration, and additional experimental analysis is necessary to clarify the molecular mechanisms underlying this phenomenon. Moreover, we can speculate on the role of different DNA double-strand break (DSB) repair pathways in the observed variations in AAV integration rates. Our findings indicate that the blunt ends generated by wtSaCas9, which are repaired predominantly through NHEJ, are more favourable for AAV integration compared to the staggered ends produced by D10ASaCas9, which are primarily repaired by MMEJ. Previous studies on natural AAV integration have suggested that it is facilitated by the non-homologous recombination pathways within the host (Yang et al, 1997; Rutledge and Russell, 1997; Daya et al, 2009). While these results are promising, it is important to note that AAV integration may have been influenced by certain experimental conditions, including the choice of the gRNA sequence or the genomic structure of the targeted region. Moreover, these conditions may not apply to other guides or regions, so further research is necessary to determine the broader applicability of the findings. Nonetheless, the results are encouraging and warrant further investigation. Interestingly, the vast majority (97.8%) of junction sites between AAV and the expected CRISPR-induced break site contained elements of the viral ITRs. However, due to the relatively short NGS read length, it was not possible to determine the length and exact rate of AAV vector genome integration events. Longer insertions that contain the full-length ITR sequences and other components of the AAV cassette may not be detectable or may be underestimated due to technical limitations resulting from the complex secondary structure of the ITR sequences, as described by Hanlon et al (2019) and Simpson et al (2023).

Regarding the safety of the paired D10ASaCas9 approach, there is a potential concern associated with the presence of highly variable insertions and deletions that could result in the production of abnormal proteins. However, it is important to note that the frequency of each specific modification is very low. Therefore, even in the unlikely event that the newly generated protein is toxic or elicits a cytotoxic immune response, the number of cells affected would be minimal, and the overall impact on the efficacy of the

treatment would be insignificant. Furthermore, CRISPR-Cas9 editing can result in unintended modifications at off-target sites as well as complex genomic rearrangements in mice, via replicative processes (Maddalo et al, 2014) with a significant impact on gene expression and genomic stability. The use of paired D10ASaCas9 should theoretically reduce, but not eliminate, the risk of off-target modifications, large genomic deletions, and other complex rearrangements that can occur during genome editing. To assess the safety of the selected guides we have employed two genome-wide evaluation methods: CIRCLE-seq (Tsai et al, 2017) and CAST-Seq (Turchiano et al, 2021). CIRCLE-seq failed to identify reliable off-target sites, as the hit showed mismatches, including the PAM. Moreover, most of the off-target hits were located in repetitive genomic regions, which introduces technical challenges for subsequent analysis. On the other hand, the CAST-Seq analysis conducted on nucleofected MEFs provided further insights. It confirmed the high editing efficacy on the target site but also identified a genomic rearrangement on chr.8. Surprisingly, this rearrangement was observed for both guides and was initially identified as an off-target-mediated translocation (OMT) only in the SaCas9g2-treated samples. However, it is important to note that Long-read Nanopore sequencing of chr.8 did not reveal any off-target editing in the edited MEFs or the liver of the treated animal. This suggests that the observed rearrangement was likely an artefact resulting from the CAST-Seq analysis conducted on MEF cells. The selected experimental conditions aimed to heighten the chance of identifying off-target effects, given the robust transfection efficiency in MEF cells and the inherent nuclease activity of Cas9. Yet, even under these stringent conditions, no off-target activity from the *Hao1*-targeting gRNAs was detected. This meant that we could not underscore the enhanced safety of the paired nickases approach, given the high specificity of the gRNAs tested. Although previous studies have already demonstrated the enhanced safety of this approach (Ran et al, 2013a; Cho et al, 2014; Shen et al, 2014), to truly validate its safety benefit, it might be necessary to repeat the experiments using guides with demonstrated off-target activity. However, it is important to note that the absence of detectable off-targets in mice might not necessarily extend to humans, making it imperative to bolster the safety of the editing system before considering clinical applications in patients.

Finally, this study employs AAV8 as an efficient delivery vector for the mouse liver. Nevertheless, it is well-established that the transduction efficiency of AAV8 in mice is significantly higher than in human hepatocytes, as evidenced by several independent studies (Lisowski et al, 2014; Cabanes-Creus et al, 2022). Thus, for this strategy to be translated to clinical applications, it would be essential to explore novel vectors with a strong affinity for human hepatocytes (Lisowski et al, 2014; Cabanes-Creus et al, 2022, 2023). Furthermore, one limitation of AAV-delivered genome editing therapies is the durability of transgene expression, potentially leading to long-term toxicities. Potential strategies to address this include the use of transient expression approaches, such as the administration of the AAV gene editing system to neonates. This approach would provide an opportunity to replenish the liver with edited hepatocytes while simultaneously eliminating vector genomes and Cas9 expression (Wang et al, 2012). If treatment in neonates is not feasible, the use of self-inactivating Cas9 might be a viable alternative (Merienne et al, 2017). In addition, the employment of nanoparticles, as proposed by Gillmore et al in 2021,

should be considered. With further optimization, nanoparticles could enable target-specific delivery of the CRISPR-Cas9 system, potentially matching the efficacy of AAV.

In summary, we have demonstrated that a single systemic administration of an all-in-one AAV-D10ASaCas9 targeting GO results in a therapeutic effect in PH1 mice. This system equals the efficiency of wtSaCas9 in vitro and in vivo and it is associated with significantly lower AAV vector integration events and absent (or undetected) off-target modifications or chromosomal translocations. To our knowledge, this is the first time that paired Cas9 nickases are used for targeted gene disruption in vivo and with a therapeutic purpose. Nevertheless, to ensure a comprehensive assessment of potential adverse events and clinical translation, further studies will be conducted using humanized models. In conclusion, our findings strongly support the efficacy and promise of employing paired D10ASaCas9-mediated GO depletion as a therapeutic strategy for PH1 patients. Continuous improvements in the future, such as incorporating self-inactivating Cas9 variants (Ibraheim et al, 2021), will enhance the potential of this approach even further.

# Methods

## Plasmid design

In order to generate the vectors encoding for D10ASaCas9 (Fig. 1A), pX602-AAV-TBG::NLS-SaCas9-NLS-HA-OLLAS-BGHpA;U6::B-saI-sgRNA plasmid a gift from Feng Zhang (Addgene plasmid # 61593; http://n2t.net/addgene:61593; RRID:Addgene_61593) (Ran et al, 2015), carrying *Hao1*-g1 described in Zabaleta et al (2018b) (*Hao1*-g1) was used. The plasmid was modified by site-directed mutagenesis (following the PCR protocol QuikChange II XL Site-Directed Mutagenesis Kit, Agilent Technologies, Part no. # 200521), introducing GAC>GCC intentional change to generate the D10ASaCas9 mutant (the primers used for site-directed mutagenesis are presented in Appendix Table S1). Thus, the plasmid pAAV-TBG::NLS-D10ASaCas9-NLS-HA-OLLAS-BGHpA-*Hao1*-g1 (TBG-D10Ag1) was generated. Then, *Hao1*-g2 was subcloned to substitute *Hao1*-g1, and TBG-D10Ag2 was generated (Fig. 1B).

To generate the plasmids described in Fig. 3A the TBG promoter was substituted with human AAT promoter and the BGH pA was substituted by a smaller synthetic pA signal. The tag present at the C-terminal end of the D10ASaCas9 (HA-OLLAS) was removed generating the AAT-D10Ag1 and AAT-D10Ag2 plasmids.

To construct the all-in-one D10A nickases system (Fig. 3A), the U6-*Hao1*-g1 cassette in the AAT-D10A-g1 plasmid was substituted by a cassette containing the two guides whose expression is controlled by U6 promoter each, generated by synthesis (GenScript), to create the AAT-D10Ag1g2 plasmid (Fig. 3A).

To generate the plasmid containing the murine *Hao1* sequence (Appendix Fig. S1A), the mouse Hao1 cDNA was amplified via PCR, using mouse liver cDNA as a template, SuperScript ® III First-Strand Synthesis System for RT-PCR (Invitrogen, Cat. No. 18080051), and oligo(dT)$_{15}$ primers (Promega, #C110A), following the manufacturer's protocol (Invitrogen) and primers (Nhe1-*Hao1* CDS Fw and Not1-*Hao1* CDS Rv; Not1-*Hao1* 3UTR-Fw and Not1-*Hao1* 3UTR Rv Appendix Table S1). Hao1 cDNA sequence was

cloned into a plasmid backbone (pIRES, Takara Bio, Cat. No. 631605), under the transcriptional control of cytomegalovirus immediate early promoter (CMV IE), using standard molecular techniques.

CMV-SaCas9 in Appendix Fig. S1A (pX601-AAV-CMV::NLS-SaCas9-NLS-3xHA-bGHpA;U6::BsaI-sgRNA) was a gift from Feng Zhang (Addgene plasmid # 61591; http://n2t.net/addgene:61591; RRID:Addgene_61591) (Ran et al, 2015). To generate CMV-SaCas9-g1 (Appendix Fig. S1A), the TBG promoter of Hao1-g1 was removed and substituted with the CMV promoter by standard molecular techniques. To generate CMV-D10A-g1g2 (Appendix Fig. S1A), the AAT promoter of AAT-D10Ag1g2 was removed and substituted with the CMV promoter by standard molecular techniques. Lastly, to generate the CMV-SaCas9-g1g2 plasmid (Appendix Fig. S1A), the D10ASaCas9 transgene was substituted with SaCas9 transgene by standard molecular techniques.

## AAV8 vector production

The plasmids described above were used to produce recombinant AAV8 vectors. HEK293T cells (ATCC CRL-3216) were co-transfected with polyethyleneimine polymer (PEI, Sigma-Aldrich #40,872-7), and the plasmids carrying the recombinant AAV genomes and a helper plasmid pDP8.ape (Plasmid Factory, #PF478). Seventy-two hours upon transfection cells and supernatant were collected. Cells were incubated in a lysis buffer (0.5 M Tris, 1.5 M NaCl, 20 mM MgCl$_2$, 1% Triton X-100) and then subjected to various cycles of freezing and thawing to release the viral particles. The supernatant was incubated with 8% polyethylene glycol (PEG-800, Sigma-Aldrich #P5413) for 48–72 h to precipitate the viral particles. The lysate of cells and the PEG-treated supernatant were treated with DNaseI (Roche, #10104159001) and RNase A (Roche, #10109169001) 0.1 mg/10 μL, for 30–60 min at 37 °C then kept at −80 °C until purification. Purification of crude lysate was performed by ultracentrifugation in Optiprep Density Gradient Medium-Iodixanol (Sigma-Aldrich). Thereafter, iodixanol was removed and the batches were concentrated by passage through Amicon Ultra-15 tubes (Ultracel-100K; Merck Millipore).

To determine vector yield, viral DNA was extracted from 20 μL of the purified vector production using the High Pure Viral DNA Kit (Roche, #11858874001). Titres of the virus were calculated by qPCR using sequence-specific primers for ITR (Appendix Table S1) and GOTaq qPCR master mix 2X (PROMEGA, #A600A) in a CFX96™ Real-Time PCR Detection system (Bio-Rad). Results were normalized with a standard curve generated using a serial dilution of the plasmid employed for the production. All reactions were performed in duplicate. AAV vectors were handled as Biosafety level 2 agents following the guidelines of the evaluation of risk committee of the Universidad de Navarra.

## Animal manipulation and procedures

Agxt1$^{-/-}$ (B6.129SvAgxttm1Ull) (Salido et al, 2006) and age-matched WT C57BL/6J (RRID:IMSR_JAX:000664) male mice were used between 7 and 14 weeks as indicated for each study. KO mice were assigned to the different experimental groups randomly. No blinding procedures were implemented in the study. $N = 2$–6, with specific numbers assigned to each group as described in the figure legend. Agxt1$^{-/-}$ mice were genotyped using KAPA HotStart Mouse Genotyping Kit (Kapa Biosystems, #KK7352) with wild-type and

mutant-specific primers together with a common forward primer, as described in Zabaleta et al (2018b).

Mice were maintained under specific pathogen-free conditions, with ad libitum access to food and water and 12-h day/night cycles. All the experimental procedures had approval by the Ethics Committee for Animal Testing of the University of Navarra (Ref. number 086-15 and ARRIVE guidelines were followed. Mice were administered with the vectors (diluted in PBS-5% sucrose P68) intravenously while under general anaesthesia (Piramal, Isofluorane Isovet 1000 mg/g). Animals that were not properly dosed were excluded for the experiment.

Animals treated with the all-in-one vector were challenged with 0.5% (v/v) EG (Sigma-Aldrich, Cat. #324558) in drinking water for 10 consecutive days. During the challenge, mice were individualized in metabolic cages for the collection of 24-h urine and monitoring of water intake. Urine samples were obtained before and on days 3 and 7 during the challenge.

Animals were sacrificed by cervical dislocation under general anaesthesia. Liver samples were collected during the necropsy. Part of the liver was fixed using 4% paraformaldehyde (PFA) for subsequent immunohistochemical analysis; the rest was frozen in liquid nitrogen for molecular analysis.

## Extraction of genomic DNA from livers and on-target PCR amplification for indel quantification analysis

Total genomic DNA was extracted from frozen liver sections using a NucleoSpin Tissue Extraction Kit (Macherey-Nagel, REF 740952.250) with a final elution in 100 μl nuclease-free H$_2$O and Qubit quantitation (Invitrogen™ Qubit™ dsDNA HS and BR Assays, Ref. #Q33230, Ref. #Q33261, respectively). On-target indel analysis was performed by deep sequencing and nested PCR was executed for NGS library preparation.

A first PCR round was performed by mixing 250 ng of template, Kapa HiFi HotStart ReadyMix PCR Kit (Roche, #KK2601), and primers (Hao1-int1 Fw and Hao1-int2 Rv; Appendix Table S1) that amplified 646 bp of the mouse Hao1 gene (containing the targeted region in the middle of the amplicon). The PCR product was purified using the Agencourt AMPure XP system (Beckman Coulter, #A63881) and eluted in 10 μl nuclease-free H$_2$O. Qubit quantitation (Invitrogen™ Qubit™ dsDNA HS and BR Assays) and TapeStation assay (Agilent Technology D1000 ScreenTape Assay for TapeStation Systems) were employed to quantify and determine the purity of the first amplicon before proceeding with the second PCR step.

Then, 40 ng of the PCR product was mixed with primers carrying partial universal adaptors for Illumina amplicons (Hao1-ex2 Fw and Hao1-ex2 Rv; Appendix Table S1). 450 bp amplicons were purified as described above and sent to GENEWIZ (GENEWIZ Germany GmbH) for Amplicon-EZ analysis.

## On-target mutagenesis analysis

Raw FASTQ files were trimmed with Trimmomatic v0.36 (Bolger et al, 2014). For the analysis of all editing events, we developed a bash script that enables the characterization of AAV insertion-bearing reads, while ensuring that these reads come from the mouse genome. First, all the reads were aligned to the mm10 reference genome with BWA MEM v0.7.17 (Li, 2013) and unmapped read-

pairs were removed from the alignment files with samtools v1.6 (Danecek et al, 2021). Then, filtered BAM files were converted back to FASTQ files with bedtools v2.30 bamToFastq (Quinlan and Hall, 2010) and were mapped with BWA MEM to the vector genome used in each experiment. Finally, samtools and bedtools were used again to only recover the reads that map to the vectors and the reads that do not map. A virus-free BAM file was obtained with minimap2 v2.17 (Li, 2018) with alignment options -A5 -B4 -O25 -E1. We analysed the editing events in the vector-sequence-free BAM files with CrispRVariants R package (Lindsay et al, 2016) and added the reads with known vector insertions to these results. To quantify the frequency of microhomology sequencing reads were analysed with mhscanr R package (Owens et al, 2019), which requires the previous execution of CRISPResso v1 (Pinello et al, 2016).

## Quantification of viral genomes

Viral genomes were quantified by qPCR using and GOTaq qPCR master mix 2X (PROMEGA, #A600A) and specific primers for *Sa*Cas9 (*Sa*Cas9 Fw and *Sa*Cas9 Rv reported in Appendix Table S1) in a CFX96™ Real-Time PCR Detection system (Bio-Rad). Relative quantification was performed using *Gapdh* as housekeeping control (specific primers for *Gapdh*, Gapdh Fw, and Gapdh Rv) are reported in Appendix Table S1. All reactions were performed in duplicate.

## RNA extraction and RT-qPCR

Total RNA from liver sections was isolated using Maxwell® RSC simplyRNA Tissue kit (Promega, #AS1340), following the protocol of the manufacturer. Total RNA was eluted in 50 μL of Nuclease-Free Water. After quantification by Nanodrop 1000 (Thermo Fisher Scientific), 1 μg of RNA was treated with TURBO DNA-free kit (Invitrogen, AM1907) to eliminate the contaminant DNA before the retro-transcription. Briefly, samples were incubated with DNase in a specific buffer at 37 °C for 30 min before DNase activity was inhibited by EDTA for 10 min at 75 °C. Next, the RNA was retrotranscribed using a mix containing RNase OUT recombinant ribonuclease inhibitor (Invitrogen, #10777019), M-MLV reverse transcriptase with DTT 0.1 M and its 5x First Strand Buffer (Invitrogen, #28025013), dNTP mix 10 mM (Invitrogen, #362275) and random primers (Invitrogen, #48190011). The reaction was a 1 h incubation at 37 °C followed by 1 min at 95 °C to inactivate the enzymes.

The expression level of SaCas9 was quantified by qPCR data analysis, using GOTaq qPCR master mix 2X (Promega, #A600A) and the specific primers listed in Appendix Table S1 (SaCas9 Fw and SaCas9 Rv). The relative amounts were calculated in relation to the Gapdh (Gapdh Fw and Rv primers listed in Appendix Table S1), according to the formula $2^{-\Delta\Delta Ct}$. All reactions were performed in duplicate.

## GO detection by western blot

Total proteins were extracted from mouse Liver or transfected cells using RIPA lysis and extraction buffer (Thermo Scientific™, Cat. no # 89900) and quantified by Pierce™ BCA Protein Assay Kit (Thermo Fisher, #23227). 20 μg of each extract was mixed with a loading buffer (containing 30% glycerol, SDS, and DTT) and boiled for up to 10 min at 95 °C. Samples were loaded in 10% polyacrylamide gels

and electrophoresis was used for protein separation by size. Subsequently, proteins were transferred to a nitrocellulose membrane (BioRad #162-0112), following the wet procedure using the BIO-RAD Trans-blot machine (membranes were stained using Ponceau Red dye to check the protein transfer). After removing the dye with water, membranes were blocked with Tris-buffered saline (TBS)-Tween 20 0.05% with non-fat dry milk 5% for 1 h at room-temperature (RT) on a shaker. Then, membranes were incubated with primary antibody α-GO (rabbit serum raised against recombinant mouse GO protein, Zabaleta et al, 2018b) in TBS-Tween 20 0.05% with non-fat dry milk 5%, at 1:5000 dilution, shaking overnight (O/N) at 4 °C. The membrane was washed three times using TBS-Tween 20 0.05% for 10 min each wash. HRP-conjugated α-rabbit IgG (GE Healthcare, #NA934V) in TBS-Tween 20 0.05% with non-fat dry milk 5%, at 1:5000 dilution was added, and membranes were incubated for 1 h at RT shaking. After three washes of 10 min in TBS-Tween 20 0.05%, the peroxidase signal was developed using SuperSignal West Femto Maximum Sensitivity Substrate (ThermoFisher Scientific, Cat. #34095) and Odyssey Fc (LI-COR) imaging system for image generation. The membrane was washed three times in TBS-Tween 20 0.05% for 10 min each wash and incubated with α-GAPDH antibody (Sigma-Aldrich, #G8795) in TBS-Tween 20 0.05% with non-fat dry milk 5%, at 1:5000 dilution shaking 1 h at RT. The membrane was washed three times using TBS-Tween 20 0.05% for 10 min each wash. HRP-conjugated α-mouse IgG (GE Healthcare, #NA931V) in TBS-Tween 20 0.05% with non-fat dry milk 5%, at 1:5000 dilution was added, and membranes were incubated for 1 h at RT shaking. GAPDH signal was detected as described above for GO. The antibodies used are summarised in Appendix Table S2.

## GO detection by immunohistochemistry

Immunohistochemical staining for GO (rabbit serum raised against recombinant mouse GO protein, Zabaleta et al, 2018b) was performed using the EnVision TM + System (Dako, Glostrup, Denmark) according to the manufacturer's recommendations. Paraffin sections (3 μm thick) were cut, dewaxed, and hydrated. Antigen retrieval was performed for 30 min at 95 °C in 0.01 M Tris-1 mM EDTA buffer (pH=9) in a Pascal pressure chamber (Dako, Code no. S2800). Slides were allowed to cool for 20 min, then endogenous peroxidase was blocked with 3% $H_2O_2$ in deionized water for 12 min and sections were washed in TBS-0.05% Tween 20 (TBS-T). Incubation with rabbit α-GO (rabbit serum raised against recombinant mouse GO protein, Appendix Table S2) at 1:2000 dilution was performed overnight at 4 °C. After rinsing in TBS-T, the sections were incubated with goat anti-rabbit (Dako, Code no. K400311-2) labelled polymer for 30 min at RT. Peroxidase activity was revealed using DAB+ (Dako, Code no. K346889-2), and sections were lightly counterstained with Harris haematoxylin. Finally, slides were dehydrated in graded series of ethanol, cleared in xylene, and mounted in Eukitt®.

## Urine oxalate/glycolate measurement

24-h urines were collected in 50 μl of 6 N HCl and their volume was measured. The urines were treated using 8% activated charcoal diluted in oxalate sample diluent (Trinity Biotech, Ref. #591-4) and urine oxalate levels were measured using the Oxalate Kit (Trinity

Biotech, Ref. #591-D) according to the manufacturer's instructions. Absorbance was measured, relative to an Oxalate standard Curve at 590 nm. Finally, oxalate μmol/24 h (μmol/24 h) was calculated using oxalate concentration and urine volume. Glycolate (μmol/24 h) was measured in charcoal-treated urine samples by gas chromatography at Reference Laboratory (reference-laboratory.es).

## CIRCLE-seq

CIRCLE-seq was performed as previously described (Tsai et al, 2017; Lazzarotto et al, 2018). Briefly, genomic DNA (gDNA) from an untreated liver sample (PH1 mice) was fragmented into 300 base pair fragments using a Covaris S2 instrument. The fragmented gDNA was end-repaired, A-tailed, and ligated with an uracil-containing stem-loop adaptor using the HTP Library Preparation Kit, PCR-free (Kapa Biosystems, Cat. no. #KK8235). DNA molecules without aptamers were then eliminated by Lambda Exonuclease (NEB, Cat. #M0262S) and *E. coli* Exonuclease I (NEB, Cat. #M0568) treatments. DNA ends were modified using the USER enzyme (NEB, Cat. #M5508) and T4 polynucleotide kinase (NEB, Cat. #M0202S). The DNA fragments were intramolecularly circularized using T4 DNA ligase (NEB, Cat. no. #M0202L), while residual linear DNA was digested with Plasmid-Safe ATP-dependent DNase (Epicentre, Cat. no. #E3110K). In vitro cleavage reactions were performed using the circularized DNA, SaCas9 protein (Synthego), and synthetic chemically modified sgRNA. The cleavage reaction was finally stopped using Proteinase K (NEB, Cat. no # P8107S), and the cleaved fragments underwent A-tailing a-tailed and ligation (as above) with a hairpin adaptor. Finally, the libraries were amplified by PCR using barcoded universal primers (NEBNext Multiplex Oligos for Illumina, NEB, Cat. no. #E7600S) and KAPA Hifi HotStart DNA polymerase (Roche, #KK2601) and sequenced by Illumina MiSeq. Sequence identity between predicted off-target loci and guide sequences was calculated accounting for degenerate bases in the PAM sequence.

## HEK293T transfection and molecular analysis

HEK293 T cells (ATCC CRL-3216) were cultivated at a density of $8 \times 10^5$ cells/well into 6 wells plate at 37 °C, 5% $CO_2$ in DMEM (1X) medium (Gibco, Ref. 41965-039) supplemented with 10% of heat-inactivated FBS (Gibco, Ref. 10500-064), 1% L-glutamine 200 mM (Corning, Ref. 25-005-CI) and 1% penicillin/streptomycin (Gibco, Ref. 15140-122). Cells were transfected at confluence with 2.5 μg/plasmids total (co-transfections at the ratio described in Appendix Fig. S1A) using Lipofectamine 3000 (Invitrogen, Ref. L3000-008) according to the manufacturer´s protocol. Transfection was performed in OPTI-MEM (Gibco, Ref. 31985-070), and 4 h post-transfection the medium of transfection was substituted with 2 ml of fresh medium. Every transfection was made in duplicate. Transfected cells were harvested 72 h post-transfection for protein extraction and expression analysis by western blot. Cells were obtained from ATCC, authenticated by SRT profile, and tested for the presence of mycoplasma contamination.

## MEF CF-1 cells and nucleofections

Mouse embryonic fibroblast (MEF) CF-1 cells (Global Stem, CF-1 MEF 4 M untreated, Catalog. Nr. GSC-6001, Lot #94300754) were cultured at 37 °C, 5% $CO_2$ in RPMI1640 medium supplemented with GlutaMAX™ (ThermoFisher, Cat. #61870010) and 10% FBS (PAN-Biotech, Cat. #P40-47500). For nucleofection, $5 \times 10^5$ MEF cells were resuspended in 20 μL of P4 solution (Lonza, Cat. #V4XP-4032) and mixed with previously assembled RNPs. For RNP assembly, 3 μg of SaCas9 protein (Synthego, SaCas9 2NLS Nuclease (300 pmol)) were complexed for 10 min with 72 pmol of gRNA (Synthego). Nucleofection was performed with a 4D-Nucleofector (Lonza, Germany), program CZ-167 in a 16-well cuvette. Cells were harvested on day 6 post nucleofection for genomic DNA extraction using the NucleoSpin Tissue Mini kit for DNA from cells and tissue (Macherey-Nagel Ref. 740952.250). Cells were obtained from Global Stem, authenticated by SRT profile, and tested for the presence of mycoplasma contamination.

## T7E1 assay and on-target PCR amplification for MEFs indel quantification analysis

T7E1 assay was performed and analysed as previously described (Dreyer et al, 2015). Briefly, 100 ng of genomic DNA was subjected to a 30-cycle PCR reaction using Q5 HotStart DNA polymerase (NEB, Cat. # M0493S) (primers 6878 and *Hao1*-int2 Rv; Appendix Table S1). 100 ng of the resulting amplicons were subjected to 5 min at 95 °C and a slow reannealing. The product was then digested with T7E1 enzyme (NEB, Cat. # M0302L) at 37 °C for 30 min and then resolved through a 2% agarose gel electrophoresis. Fragments were quantified using ImageJ software (ImageJ 1.47v). In addition, 100 ng of genomic DNA was subjected to a 30-cycle PCR reaction using Q5 HotStart DNA polymerase (primers 7783 and 7784; Appendix Table S1) and sent to GENEWIZ (GENEWIZ Germany GmbH) for Amplicon-EZ analysis.

## CAST-Seq

CAST-Seq was performed as previously described (Turchiano et al, 2021) with minor modifications. Briefly, genomic DNA was fragmented by enzymatic digestion (NEBNext® Ultra™ II FS DNA Library Prep Cat. # E7805S) to obtain average fragment lengths of 350–500 bp. After linker ligation (NEBNext® Ultra™ II FS DNA Library Prep Cat. # E7805S) and DNA purification, two rounds of PCR utilizing Q5 polymerase (NEB, Cat. # M0493S) were performed with the following conditions: 20 cycles at 95 °C for 15 s, 61°/63 °C (g1 or g1 + 2/g2, first reaction) or 68 °C (second reaction) for 20 s, 72 °C for 20 s. A third PCR introduced the barcoded Illumina adaptor for sequencing (NEB, NEBNext Multiplex Oligos for Illumina, Cat. # E7335, E7500, E7710, E7730). Libraries were sequenced by Azenta Life Sciences using the Illumina NovaSeq platform with a $2 \times 150$ bp configuration. Oligonucleotide sequences are reported in Appendix Table S1. The bioinformatic analysis was performed with an improved classification and scoring of the identified translocation reads (Rhiel et al, 2023).

## Long-read Nanopore sequencing

200 ng of genomic DNA from MEF cells or mouse liver samples were used as input for a 30-cycle PCR amplification using KAPA Hifi HotStart DNA polymerase (Roche, #KK2601) (7659 Fw1, 7663 Rv1, 7660 Fw2, 7664 Rv2, 7661 Fw3, 7665 Rv3, 7662 Fw4, 7666 Rv4 primer sequences are reported in Appendix Table S1). PCR products were

**The paper explained**

**Problem**

PH1 is a rare genetic disease characterized by the overproduction of oxalate in the liver, leading to its toxic accumulation in the kidneys and eventually end-stage renal disease and death. The only curative treatment is liver-kidney transplantation. Current therapeutic approaches involve reducing the expression of glycolate oxidase (GO) enzyme through RNA interference. However, a permanent reduction in GO expression is highly desirable. CRISPR-Cas9 nuclease is the most promising tool for long-lasting gene disruption, although limited by the risk of unintended genetic alterations.

**Results**

In our study, we used paired *Staphylococcus aureus* Cas9 nickases (D10ASaCas9) to disrupt the *Hao1* gene and permanently reduce GO expression in PH1 mice. Nickases offer improved accuracy by requiring two guides to mediate double-strand breaks, thus reducing the risk of off-target modifications. We delivered the nickases using adeno-associated viral vectors (AAV) and observed a significant and permanent decrease in GO expression as well as therapeutic efficacy in PH1 mice. Conversely and as expected, single nicks failed to disrupt the target gene, supporting the safety of the strategy. Unexpectedly, we found that the use of paired Cas9 nickases significantly reduced the frequency of AAV genome integration events compared to Cas9 nuclease. Moreover, safety assessments showed no off-target activity or chromosomal translocations.

**Impact**

The significance of our study lies not only in its potential to provide a permanent solution for PH1 patients but also in the innovative use of paired Cas9 nickases to enhance the accuracy of CRISPR-Cas9 gene disruption, which might have important implications for the broader field of gene therapy.

purified using AMPure beads with a 1.8x ratio of beads to DNA, and libraries were prepared using the Nanopore Ligation Sequencing Kit V14 (SQK-LSK114) and associated protocol. The samples were loaded and sequenced on a MinION Flow Cell. Reads were demultiplexed according to the barcodes used for each sample using Last V.6 (Eccles, 2022). Obtained reads were aligned using Minimap2 –ax splice v. 2.24-r1122 (Li, 2018), further processed with Samtools (Danecek et al, 2021) v. 1.9, and plotted using IGV (Robinson et al, 2011). The deletion read coverage was normalized to the total amount of reads covering each position of the amplicon, mean and standard deviation were calculated.

## Statistics

We used GraphPad Prism v.9.0 and SciPy v.1.10 for statistical analysis. To compare means, we used an unpaired two-tailed t-test (after Shapiro–Wilk normality testing); to compare multiple groups, we used analysis of variance (one-way ANOVA) followed by Tukey's post-hoc test. For the experiment with multiple time points, we used two-way ANOVA followed by Tukey's post-hoc test. $P$ values < 0.05 were accepted as significant.

## Data availability

The datasets and computer code produced in this study are available in the following databases: Targeted amplicon sequencing data: NCBI

Sequence Read Archive PRJNA966811. Scripts for amplicon sequencing analysis: GitHub (https://github.com/bilbaom/ampseq-aav-insertions). CAST-Seq data: NCBI GEO GSE245791.

## Peer review information

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

## Acknowledgements

We thank the animal facility staff at CIMA, the Genomics Facility at CIMA-LAB Diagnostics, and the CIMA-Morphology Platform. A sincere thank you to Dr. M. Hommel and Dr. N. Weber for their diligent proofreading of the manuscript and Dr C. Gázquez for the technical assistance. We also sincerely acknowledge the following institutions: Spanish Agencia Estatal de Investigación grant RTI2018-101936-B-I00 (to GGA); Spanish Network of Advanced Therapies TERAV Network supported by Instituto de Salud Carlos III (ISCIII) and Funded by the European Union – NextGenerationEU, Recovery, Transformation and Resilience Plan grant RD21/0017/0001 (to GGA and MBA); Centro para la Investigación Médica Aplicada (CIMA) fellowship (to LT); Oxalosis and Hyperoxaluria Foundation (OHF) fellowship (to LT); AAVolution - Next-Generation AAV Vectors for Liver-Directed Gene Therapy EC project number 101071041 (to GGA and LT); GenE-HumDi Genome Editing for the treatment of human Disease CA21113 (to LT, MBA, TC and NZ); German Research Foundation grant CA311/4-1 and CRC1160 (Project ID 256073931- A07 to TC and Z02 to MB), CRC/TRR167 (Project ID 259373024-Z01), CRC1453 (Project ID 431984000-S1), and CRC1479 (Project ID: 441891347- S1) (to MB). German Federal Ministry of Education and Research (BMBF) grant MIRACUM-FKZ 01ZZ1801B (to MB); German Federal Ministry of Education and Research (BMBF) grant EkoEstMed–FKZ 01ZZ2015 (to GA).

## Author contributions

**Laura Torella**: Conceptualization; Formal analysis; Investigation; Methodology; Writing—original draft. **Julia Klermund**: Investigation; Methodology; Writing—original draft. **Martin Bilbao-Arribas**: Conceptualization; Software; Formal analysis; Investigation; Methodology; Writing—original draft. **Ibon Tamayo**: Software; Validation; Investigation; Methodology. **Geoffroy Andreiux**: Data curation; Software; Validation. **Kay O Chmielewski**: Data curation; Formal analysis; Validation; Methodology. **Africa Vales**: Methodology; Project administration. **Cristina Olagüe**: Methodology. **Daniel Moreno-Luqui**: Investigation; Methodology. **Ivan Raimondi**: Data curation; Formal analysis; Investigation; Methodology. **Amaya Abad**: Methodology. **Julen Torrens-Baile**: Software; Formal analysis; Methodology. **Eduardo Salido**: Conceptualization; Writing—review and editing. **Maite Huarte**: Methodology; Writing—review and editing. **Mikel Hernaez**: Software; Funding acquisition; Validation; Methodology. **Melanie Börries**: Software; Formal analysis; Supervision; Funding acquisition. **Toni Cathomen**: Conceptualization; Resources; Formal analysis; Supervision; Funding acquisition; Investigation; Writing—review and editing. **Nerea Zabaleta**: Conceptualization; Supervision; Investigation; Methodology; Writing—original draft; Writing—review and editing.
**Gloria Gonzalez-Aseguinolaza**: Conceptualization; Formal analysis; Supervision; Funding acquisition; Writing—original draft; Writing—review and editing.

## Disclosure and competing interests statement

ES holds shares of Orfan Biotech. TC is a consultant to Cimeio Therapeutics and Excision BioTherapeutics, and holds a patent on CAST-Seq. GG-A is a founder and shareholder of Vivet Therapeutics. All other authors declare they have no competing interests.

# Expanded View Figures

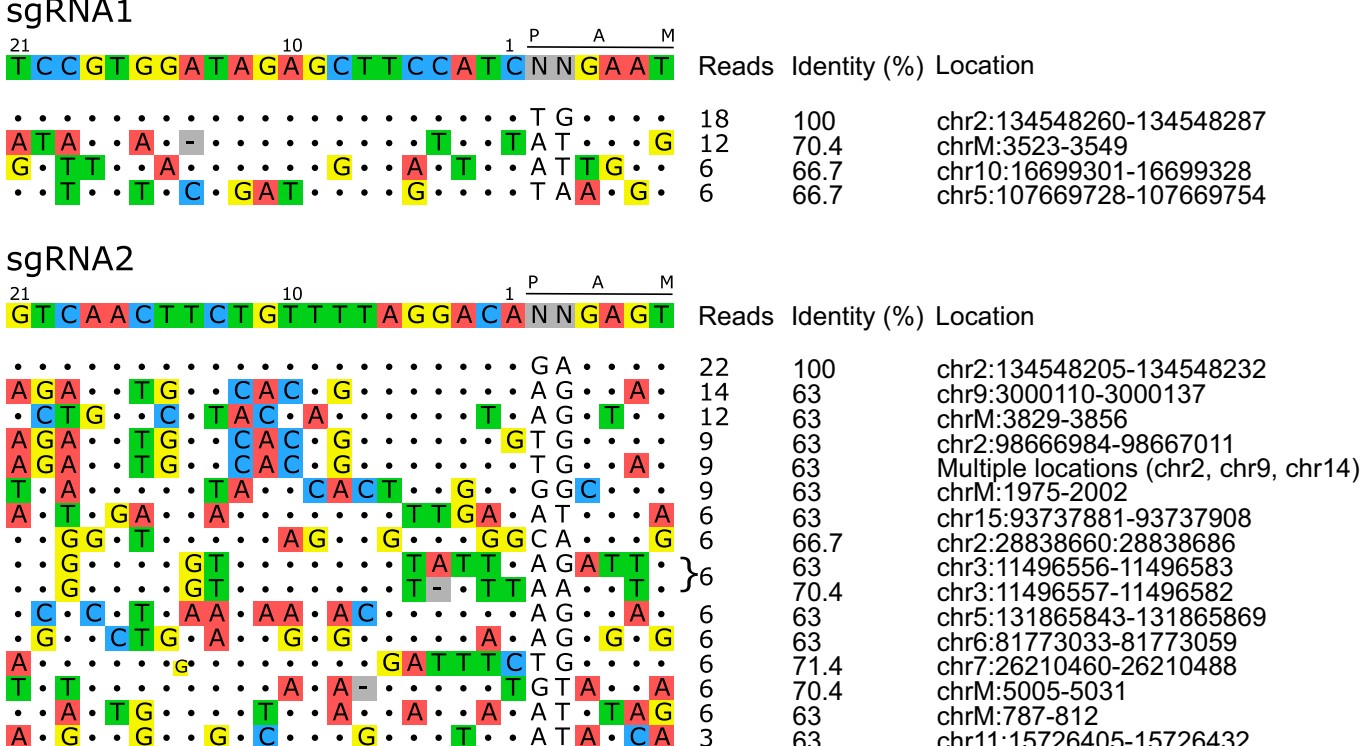

**Figure EV1. CIRCLE-seq screening for CRISPR-off-target analysis.**

Representation of the off-targets identified by CIRCLE-seq aligned against the intended target site for SaCas9:*Hao1*-g1 and SaCas9:*Hao1*-g2 targeting the *Hao1* gene. The intended target sequence is shown in the top line, the on-target is the first appearing in the below list and the off-target sites are ordered from top to bottom by CIRCLE-seq read count. Perfect base matches are represented as dots while mismatches to the intended target sequence are indicated by coloured nucleotides. Read counts are shown at the end of each line, on the right. Identity of the intended target sequence was calculated and shown at the end of each line, on the right. Chromosomal coordinates for each off-target site are shown at the end of each line, on the right.

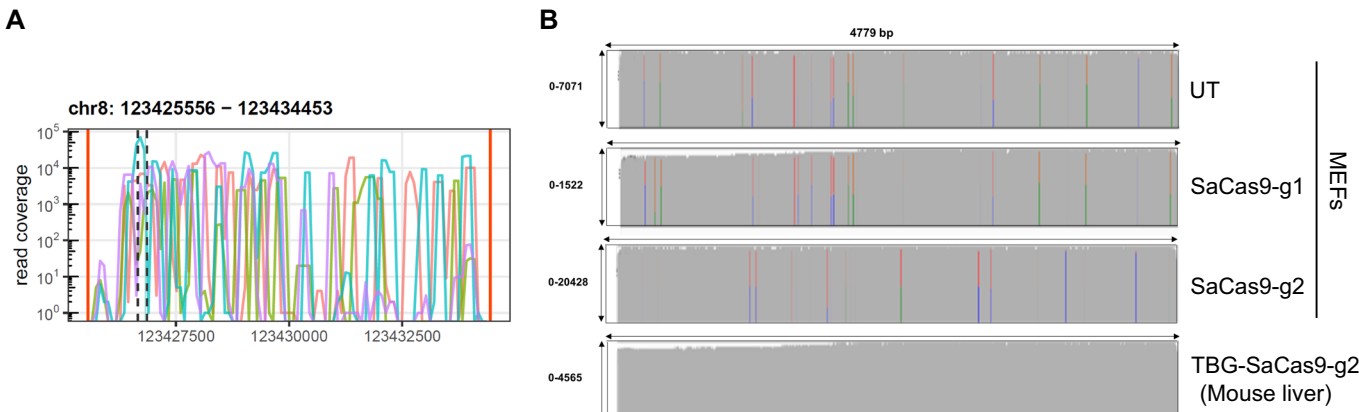

**Figure EV2. Targeted long-read sequencing of the OMT detected by CAST-Seq demonstrates the specificity of the selected gRNAs.**

(A) Coverage plot from the CAST-Seq result showing which reads from chr.8 were translocated onto the on-target site. (B) Read coverage of the long-read sequencing spanning 5 kb of chr.8 of untreated MEF (UT-MEF), electroporated MEFs (SaCas9g1-MEF and SaCas9g2-MEF), and a liver sample of PH1 mouse treated with TBG-SaCas9g2. $N = 1$ biological replicate per condition is shown. The deletion read coverage was normalized to the total amount of reads covering each position of the amplicon (mean ± SD).

