## [Peer Review File · EMBO Molecular Medicine]

Efficient and Safe Therapeutic Use of Paired Cas9-Nickases for Primary Hyperoxaluria Type 1

Laura Torella, Julia Klermund, Martin Bilbao-Arribas, Ibon Tamayo, Geoffroy Andreiux, Kay Chmielewski, Africa Vales, Cristina Olagüe, Daniel Moreno-Luqui, Ivan Raimondi, Amaya Abad, Julen Torrens-Baile, Eduardo Salido, Maite Huarte, Mikel Hernaez, Melanie Börries, Toni Cathomen, Nerea Zabaleta, and Gloria Gonzalez-Asequinolaza

DOI: [10.15252/emmm.202318234](https://doi.org/10.15252/emmm.202318234)

Corresponding authors: Gloria Gonzalez-Asequinolaza (ggasegui@unav.es) , Toni Cathomen (toni.cathomen@uniklinik-freiburg.de)

Review Timeline:

Submission Date:	5th Jul 23
Editorial Decision:	4th Aug 23
Revision Received:	25th Oct 23
Editorial Decision:	7th Nov 23
Revision Received:	14th Nov 23
Accepted:	15th Nov 23

Editor: Lise Roth

Transaction Report:

4th Aug 2023

Dear Dr. Gonzalez-Aseguinolaza,

Thank you for the submission of your manuscript to EMBO Molecular Medicine. We have now received feedback from the three reviewers who agreed to evaluate your manuscript. As you will see from the reports below, the referees acknowledge the interest of the study and are overall supporting publication of your work pending appropriate revisions.

Addressing the reviewers' concerns in full will be necessary for further considering the manuscript in our journal, and acceptance of the manuscript will entail a second round of review. EMBO Molecular Medicine encourages a single round of revision only and therefore, acceptance or rejection of the manuscript will depend on the completeness of your responses included in the next, final version of the manuscript. For this reason, and to save you from any frustrations in the end, I would strongly advise against returning an incomplete revision.

We are expecting your revised manuscript within three months, if you anticipate any delay, please contact us.

We require:

4) A .docx formatted letter INCLUDING the reviewers' reports and your detailed point-by-point responses to their comments. As part of the EMBO Press transparent editorial process, the point-by-point response is part of the Review Process File (RPF), which will be published alongside your paper.

5) A complete author checklist, which you can download from our author guidelines (<https://www.embopress.org/page/journal/17574684/authorguide#submissionofrevisions>). Please insert information in the checklist that is also reflected in the manuscript. The completed author checklist will also be part of the RPF.

6) Please note that all corresponding authors are required to supply an ORCID ID for their name upon submission of a revised manuscript.

7) It is mandatory to include a 'Data Availability' section after the Materials and Methods. Before submitting your revision, primary datasets produced in this study need to be deposited in an appropriate public database, and the accession numbers and database listed under 'Data Availability'. Please remember to provide a reviewer password if the datasets are not yet public (see <https://www.embopress.org/page/journal/17574684/authorguide#dataavailability>).

8) For data quantification: please specify the name of the statistical test used to generate error bars and P values, the number (n) of independent experiments (specify technical or biological replicates) underlying each data point and the test used to calculate p-values in each figure legend. The figure legends should contain a basic description of n, P and the test applied. Graphs must include a description of the bars and the error bars (s.d., s.e.m.). Please provide exact p values.

9) Our journal encourages inclusion of *data citations in the reference list* to directly cite datasets that were re-used and obtained from public databases. Data citations in the article text are distinct from normal bibliographical citations and should directly link to the database records from which the data can be accessed. In the main text, data citations are formatted as follows: "Data ref: Smith et al, 2001" or "Data ref: NCBI Sequence Read Archive PRJNA342805, 2017". In the Reference list,

data citations must be labeled with "[DATASET]". A data reference must provide the database name, accession number/identifiers and a resolvable link to the landing page from which the data can be accessed at the end of the reference. Further instructions are available at .

13) Author contributions: CRediT has replaced the traditional author contributions section because it offers a systematic machine readable author contributions format that allows for more effective research assessment. Please remove the Authors Contributions from the manuscript and use the free text boxes beneath each contributing author's name in our system to add specific details on the author's contribution. More information is available in our guide to authors.

16) As part of the EMBO Publications transparent editorial process initiative (see our Editorial at <http://embomolmed.embopress.org/content/2/9/329>), EMBO Molecular Medicine will publish online a Review Process File (RPF) to accompany accepted manuscripts.

In the event of acceptance, this file will be published in conjunction with your paper and will include the anonymous referee reports, your point-by-point response and all pertinent correspondence relating to the manuscript. Let us know whether you agree with the publication of the RPF and as here, if you want to remove or not any figures from it prior to publication.

I look forward to receiving your revised manuscript.

Yours sincerely,

Lise Roth

***** Reviewer's comments *****

Referee #1 (Comments on Novelty/Model System for Author):

The quality of the work is high and while the approach is new, it should be admitted that there are already many similar approaches to PH1. Also, the AAV platform is highly valuable for gene addition, although for cas9 mediated gene disruption one would think that a more transient delivery method would be more appropriate.

Referee #1 (Remarks for Author):

The manuscript by Torella and colleagues describes a new Cas9 system for the disruption of the Glycolate Oxidase (GO) gene to rescue Primary Hyperoxaluria type 1 (PH1). The authors previously showed that they could use a *Staphylococcus aureus* Cas9 (SaCas9) nuclease with a single gRNA or two AAV vectors with two different gRNA to disrupt expression of the GO gene and rescue the disease.

Here, using a similar delivery approach but with a *Staphylococcus aureus* nickase (D10ASaCas9) with two gRNA, the authors show that they could achieve similar therapeutic efficacy with reduced AAV vector integration, low to no off targets and no detectable chromosomal translocations.

The work is exciting, innovative and interesting, as it advances the Cas9 mediated platform by using variants better supporting the safe translation of results to the clinic.

Main comments:

The authors put some emphasis on the fact that using a single AAV construct with D10ASaCas9 and the two gRNA is an approach superior to using two AAV vectors expressing D10ASaCas9 and each gRNA. This based on AAV integration data and efficacy of GO disruption. It is not entirely clear whether the data support the conclusions as the two system appear to yield very similar results. Also, the authors claim that the "difference" in efficacy is driven by the co-expression of the two gRNA in the same cells (vs. having two vectors each expressing one gRNA). However, at the doses tested in mice it seems likely that the vectors are co-expressed in every cells anyway.

What would drive the lower integration of the AAV in mice treated with vectors carrying the AAT promoter vs. TBG?

When thinking about the clinical translation of the approach, one should keep in mind that 1. The efficacy of transduction of hepatocytes in mice is far higher than NHPs or humans; 2. The expression deriving from AAV vectors last for a long time, while the ideal for the proposed therapeutic modality is a delivery system that allows for a hit and run approach. This should be clarified in the discussion.

Referee #2 (Remarks for Author):

SUMMARY

In this work, Torella et al. have developed and characterized a dual-nicking (i.e. paired nickase) SaCas9 (D10A) based gene editing strategy to disrupt the *Hao1* gene in a mouse model of Primary Hyperoxaluria type 1 (PH1). The authors demonstrate efficient gene disruption and therapeutic effects in the preclinical model system. Importantly, the authors perform extensive analyses to detect potential (1) on-target/bystander editing e.g., chromosomal re-arrangements, (2) genome-wide off-target effects, as well as (3) on-target AAV integration. While (1) and (2) are not detectable, (3) is substantially reduced compared to an analogous nuclease approach.

ASSESSMENT

This work is extremely interesting for the genomic medicine and gene editing communities and is of excellent quality. Using a dual-nicking approach in such a therapeutically relevant setting (in vivo!) is exciting per se. Obtaining efficient editing outcomes with therapeutic efficacy together with dramatically reduced AAV integrations and undetectable off-targets/rearrangements is even more impressive. Furthermore, the single-AAV (all-in-one) vector solution that the authors developed allows a substantial reduction of the AAV dose. The findings on the dosage and efficiency of single vs. dual-AAV strategies are important for AAV-based delivery per se, particularly in the context of newer CRISPR 2.0 technologies. While no definitive mechanistic proof is shown, the findings and speculations on NHEJ-mediated AAV integration upon blunt cutting vs. MMEJ-mediated repair upon staggered cutting might have important implications for other classes of CRISPR tools as well, especially given the growing popularity of size-reduced, staggered cutting type V systems. Some minor comments should be addressed (see below), but all in all, this is a highly relevant and well-made manuscript. Congrats to the authors!

MINOR

1. Would the authors expect the "opposite" SaCas9 nickase (N580A or H557A) to work comparably to D10A? What was the rationale for choosing D10A for the dual-nicking approach? Might be helpful to discuss different nick-to-nick distances, too.
 2. Line 127/128: In this passage it was not entirely clear to me why the dual nicking approach was chosen or switched to (from nuclease), in the first place. Reducing off-targets makes sense, but it reads here as if you detected substantial off-targets with the nuclease approach and therefore wanted to switch. Was that the case? If so, it would be great to describe in more detail what the off-targets looked like or how abundant they were with the nuclease approach.
 3. Line 135: I think it would make sense to refer to SaCas9 nickase specifically here because there are some differences between e.g., SpCas9 and SaCas9 when it comes to mutations of the HNH domain. Therefore, I wouldn't necessarily mention N863A here, given that this was an initial HNH mutant for SpCas9, and H840A is by far the most commonly used mutant for this purpose nowadays. One might consider mentioning the alternative SaCas9 HNH nicking variant H557A though (PMID 26524662, 26317473).
 4. Line 136: I think the authors swapped RuvC and HNH here.
D10 = RuvC, N580 = HNH.
- Friedland et al., Genome Biology 2015: „We used site-directed mutagenesis to generate D10A and N580A mutants that would similarly disable the RuvC and HNH nuclease domains, respectively.“
5. Line 194: What approach has higher vector dosing, nuclease or (current) dual-nick? Here, it sounds as if dual-nicking approach has lower dose but in line 233 it sounds like it's the other way around.
 6. Line 233: Could the authors please briefly explain why a 5x higher dose was needed for the dual-nicking approach compared to the WTSaCas9 nuclease?
 7. Line 373: It would be helpful to explain why the translocation was detected in all experimental conditions but only classified as an OMT in the g2 treatment.
 8. Fig. 1a: I would give the PAMs a little "PAM" label. Also, cleavage is indicated at 2bp upstream of the PAM but probably is expected 3bp upstream for SaCas9, see Extended Data Fig. 2 of Ran et al, Nature 2015 (PMID 25830891).
 9. Fig. 1b: Is there a specific reason why the D10A-SaCas9 is thicker in the upper construct?
 10. Fig. 2a/b: What do the labels left of the graph mean, e.g., 30:23D? Also, would it be possible to add allele frequencies here?

Referee #3 (Comments on Novelty/Model System for Author):

The mouse disease model was established well.

Referee #3 (Remarks for Author):

In this study, Torella et al. explored the application of AAV-delivered paired SaCas9 nickase to disrupt the Hao1 gene to treat

primary hyperoxaluria type 1 (PH1). They showed that using an all-in-one AAV system to deliver SaCas9 D10A nickase and dual sgRNAs could effectively reduce the expression of glycolate oxidase (GO) enzyme, thus bringing about an obvious therapeutic effect in the mouse disease model. Importantly, this nickase strategy outperformed the conventional nuclease method in a significant reduction in AAV integration at the target site, which may be caused by different repair pathways determined by cutting ends. In general, this work provides important evidence showing the advantage of SaCas9 nickase approach, proposing an improved method for in vivo gene therapy. Despite the potential value, a part of current data is not solid enough to support the conclusions. More experiments should be performed before publication.

Major points:

- (1) Line 195-198: As the editing efficiency of paired nickase method is a big concern, the authors did not strictly compare the efficiency of single, paired wild-type SaCas9 and paired SaCas9 nickase in one experiment. Since conditions vary between different experiments, comparing the current data with the previous ones is not reasonable.
- (2) Although the authors carried out CIRCLE-seq and CAST-seq to detect potential off-target events and chromosomal translocations, the assays were performed in MEFs, which was irrelevant with the treatment strategy that the study proposed. Given that off-target activities are always determined by specific conditions, an in vivo detection assay is needed. Also, importantly, in order to conclude the low frequency of off-target events when using paired SaCas9 nickase, the authors must compare it with wild-type SaCas9 with one or two sgRNAs in parallel.
- (3) Since the significance of this study is to demonstrate paired SaCas9 nickase is a superior method for gene therapy, data of targeting more sites besides current two in the Hao1 gene, showing comparable efficiency and reduced unintended AAV integration and off-target events, are indispensable. Moreover, considering this study only provided the evidence of effective treating PH1 by using paired SaCas9 nickase, it seems to be more proper to write "treating PH1" rather than "showing gene disruption" in the title.

Minor points:

- (1) In figure 1, the group setting is not consistent among figure 1C, 1D and 1E.
- (2) Line 233: The authors mentioned "TBG-D10Ag1+g2-treated mice received five times more AAV vector genomes than those treated with TBG-SaCas9g1+g2", but detailed experimental conditions, such as dose and administration time, are deficient in the current sections of "results" and "materials and methods".
- (3) In figure 3B, the expression of GO looks higher in the high dose group than that in the median or low dose group, which is inconsistent with the result in figure 3C. Could the authors give an explanation?
- (4) Line 352: It is somewhat unclear for readers to know which off-target site is described by the authors in figure EV8. It will be helpful to add chromosome location for each off-target site.

Answer to the referees**Point-by-point response**

Referee #1 (Comments on Novelty/Model System for Author):

The quality of the work is high and while the approach is new, it should be admitted that there are already many similar approaches to PH1. Also, the AAV platform is highly valuable for gene addition, although for cas9 mediated gene disruption one would think that a more transient delivery method would be more appropriate.

We agree with the referee on the benefit of using a transient delivery method such as lipid nanoparticles containing RNA. However, the use of AAV vectors offers advantages such as a very high transduction efficiency of the target cells and the cell-specific expression of the nuclease thanks to the use of cell-type-specific promoters. This said, we acknowledge that transient delivery methods will be ideal when the other technologies have been further developed. This point has now been discussed in the discussion section. Lines: 519-529.

The following paragraph has been added (lines: *Furthermore, one limitation of AAV-delivered genome editing therapies is the durability of the transgene expression, potentially leading to long-term toxicities. Potential strategies to address this include the use of transient expression approaches, such as the administration of the AAV gene editing system to neonates. This approach would provide an opportunity to replenish the liver with edited hepatocytes while simultaneously eliminating vector genomes and Cas9 expression (Wang et al, 2012). If treatment in neonates is not feasible, the use of self-inactivating Cas9 might be a viable alternative (Merienne et al, 2017). Additionally, the employment of nanoparticles, as proposed by Gillmore et al. in 2021, should be considered. With further optimization, nanoparticles could enable target-specific delivery of the CRISPR-Cas9 system, potentially matching the efficacy of AAV.*

Wang L, Wang H, Bell P, McMenamin D & Wilson JM (2012) Hepatic gene transfer in neonatal mice by adeno-associated virus serotype 8 vector. *Hum Gene Ther* 23: 533–539

Merienne N, Vachey G, de Longprez L, Meunier C, Zimmer V, Perriard G, Canales M, Mathias A, Herrgott L, Beltraminelli T, et al (2017) The Self-Inactivating KamiCas9 System for the Editing of CNS Disease Genes. *Cell Rep* 20: 2980–2991

Referee #1 (Remarks for Author):

The manuscript by Torella and colleagues describes a new Cas9 system for the disruption of the Glycolate Oxidase (GO) gene to rescue Primary Hyperoxaluria type 1 (PH1). The authors previously showed that they could use a *Staphylococcus aureus* Cas9 (SaCas9) nuclease with a single gRNA or two AAV vectors with two different gRNA to disrupt expression of the GO gene and rescue the disease.

Here, using a similar delivery approach but with a *Staphylococcus aureus* nickase (D10ASaCas9) with two gRNA, the authors show that they could achieve similar therapeutic efficacy with reduced AAV vector integration, low to no off targets and no detectable chromosomal translocations.

The work is exciting, innovative and interesting, as it advances the Cas9 mediated platform by using variants better supporting the safe translation of results to the clinic.

Main comments:

The authors put some emphasis on the fact that using a single AAV construct with D10ASaCas9 and the two gRNA is an approach superior to using two AAV vectors expressing D10ASaCas9 and each gRNA. This based on AAV integration data and efficacy of GO disruption. It is not entirely clear whether the data support the conclusions as the two system appear to yield very similar results. Also, the authors claim that the "difference" in efficacy is driven by the co-expression of the two gRNA in the same cells (vs. having two vectors each expressing one gRNA). However, at the doses tested in mice it seems likely that the vectors are co-expressed in every cells anyway.

We thank the referee for the comments. Our conclusions are mainly derived from Figure 3 and the data obtained when using the low vector dose (5×10^{12} vg/kg) of all-in-one vector or the two independent vectors. In this study, we observed that the total number of AAV genomes in the liver of the two groups is the same, thus the differences can not be attributed to the administration of a higher dose of the all-in-one vector. The animals treated with the all-in-one vector showed no or very low levels of the GO protein in the liver, and the mean editing frequency on target in the 4 animals analyzed is 54.45%. However, GO expression is clearly detectable in the animals receiving the same dose of the two independent vectors and the mean editing frequency analyzed in 3 of these animals is 37.8%. All in all, our hypothesis for the reduced effectiveness of the two independent vectors is that certain hepatocytes might have only received one vector containing a single gRNA or the amount of each gRNA in the cell might not be equal and this imbalance might prevent from having enough of both gRNAs for simultaneous nicking. These comments have now been included in the discussion. Lines: 406-414

What would drive the lower integration of the AAV in mice treated with vectors carrying the AAT promoter vs. TBG?

First, we re-evaluated the number of vector genomes in the liver of mice treated with the TBG and the AAT vectors at the same dose using qPCR. For this analysis, we used newly extracted DNA, and all samples were analyzed simultaneously. As depicted in appendix Figure S9, we noted fewer AAV genome copies and reduced Cas9 expression levels (although statistically not significant) in the group treated with the AAT vector compared to the group that received the TBG vector (1.6-fold difference). This variance in the quantity of vector genomes could account for the 2-fold difference observed in AAV vector integration. In light of these findings, we have updated our description of the results. (Lines: 297-300).

When thinking about the clinical translation of the approach, one should keep in mind that 1. The efficacy of transduction of hepatocytes in mice is far higher than NHPs or humans; 2. The expression deriving from AAV vectors last for a long time, while the ideal for the proposed therapeutic modality is a delivery system that allows for a hit and run approach. This should be clarified in the discussion.

We have introduced a comment in the discussion acknowledging the difference on AAV-mediated liver transduction in mice vs NHPs or humans and highlighted the need of using more efficient vector in patients that are currently under development. (Lines: 514-519).

See the answer to the referee's first comment.

Referee #2 (Remarks for Author):

SUMMARY

In this work, Torella et al. have developed and characterized a dual-nicking (i.e. paired nickase) SaCas9 (D10A) based gene editing strategy to disrupt the *Hao1* gene in a mouse model of Primary Hyperoxaluria type 1 (PH1). The authors demonstrate efficient gene disruption and therapeutic effects in the preclinical model system. Importantly, the authors perform extensive analyses to detect potential (1) on-target/bystander editing e.g., chromosomal re-arrangements, (2) genome-wide off-target effects, as well as (3) on-target AAV integration. While (1) and (2) are not detectable, (3) is substantially reduced compared to an analogous nuclease approach.

ASSESSMENT

This work is extremely interesting for the genomic medicine and gene editing communities and is of excellent quality. Using a dual-nicking approach in such a therapeutically relevant setting (in vivo!) is exciting per se. Obtaining efficient editing outcomes with therapeutic efficacy together with dramatically reduced AAV integrations and undetectable off-targets/rearrangements is even more impressive. Furthermore, the single-AAV (all-in-one) vector solution that the authors developed allows a substantial reduction of the AAV dose. The findings on the dosage and efficiency of single vs. dual-AAV strategies are important for AAV-based delivery per se, particularly in the context of newer CRISPR 2.0 technologies. While no definitive mechanistic proof is shown, the findings and speculations on NHEJ-mediated AAV integration upon blunt cutting vs. MMEJ-mediated repair upon staggered cutting might have important implications for other classes of CRISPR tools as well, especially given the growing popularity of size-reduced, staggered cutting type V systems. Some minor comments should be addressed (see below), but all in all, this is a highly relevant and well-made manuscript. Congrats to the authors!

We are very grateful to the referee for the enthusiastic comments and support for publication.

MINOR

1. Would the authors expect the "opposite" SaCas9 nickase (N580A or H557A) to work comparably to D10A? What was the rationale for choosing D10A for the dual-nicking approach? Might be helpful to discuss different nick-to-nick distances, too.

This is an excellent question. Not many original manuscripts discuss the use of a paired Cas9 nickases approach, and even fewer address the use of paired SaCas9 nickases and the nature of the lesions and repair pathways activated *in vivo*. However, a study conducted by Bothmer *et al* (2017) in U2OS cells demonstrates that the presence and polarity of the overhang structure play a critical role in determining the double-strand break repair pathway chosen. In their research, using a pair of guides targeting opposite strands of the hemoglobin beta (HBB) locus at a distance of 47 bp with PAMs facing outward relative to each other (very similar to the design used in our study), they compared the editing efficacy and the types of DNA lesions introduced by D10A and N863A SpCas9. They reported a higher efficacy for D10A SpCas9 compared to N863A SpCas9. What was even more striking were the differences in the repair outcomes. Bothmer et al. found that paired D10A, resulting in DSBs with 5' overhangs, mainly induced gene conversion and deletions, while N863A, resulting in DSBs with 3' overhangs, primarily caused insertions and deletions but showed very low gene conversion rates. The gene conversion observed in their study resulted from the activation of the homologous recombination of the HBB locus

with the homologous delta gene, indicating that D10A-induced 5' overhangs favor the activation of the homologous recombination repair pathway (HDR). Additional studies using different guides and locus allowed the authors to conclude that D10A Cas9 favors HDR while N863 Cas9 favors alternative Non homologous end joining (a-NHEJ). Therefore, Bothmer *et al*'s findings suggest that the generation of different end structures may engage distinct repair pathways.

Based on these observations, we are left with the open question of whether the use of N580A or H557A SaCas9 might produce different results from those obtained with D10A due to the generation of overhangs with different polarity, which will trigger a different repair mechanism. However, it is important to take into account that HDR requires that the cell enters the G2/S phase and adult hepatocytes (our target cell) are predominantly quiescent in G0 phase. This quiescent state likely accounts for the prevalence of a-NHEJ, as we observed. Additionally, it may also lead to a comparable outcome following cleavage with D10A or N580A/H557A.

Thus, the short answer will be that we will anticipate similar results with N580A or H557A and D10A SaCas9 nickase in terms of the characteristics of genome modification.

However, experiments comparing the N580A SaCas9 nickase and the D10A will be initiated soon to ensure that the optimal version of SaCas9 is chosen for this therapy.

The selection of the D10A SaCas9 was based on the *in vitro* data presented by Friedland *et al*, 2015. In that study paired D10A SaCas9 outperformed paired N580A when targeting *VEGFA* locus at the different sgRNA distances investigated. The same is true in the work previously mentioned by Bothmer *et al*, 2017 showing the superiority of D10A but in this case SpCas9. In addition, recent reports indicate that the D10A mutation in SpCas9 has nickase activity with no tendency to generate DSBs, while other nickase versions, such as H840A tend to generate DSBs in the target site in the presence of a single nick (Lee J *et al*, 2023). This further supports the use of D10A in our strategy.

Regarding different nick-to-nick distances we mention the paper by Friedland *et al*. 2015 in which the show that the gRNAs distance can profoundly influence indel rates. According to this study the distance of our guides, which is 64 bp, falls within the optimal range for efficient indel formation, which spans from 0 to 170 bp.

We have included in the discussion the potential impact over indel frequency and distribution of using alternative Cas9 variants and the use of guides with different offset distances. Lines: 385-389.

Bothmer A, Phadke T, Barrera LA, Margulies CM, Lee CS, Buquicchio F, Moss S, Abdulkarim HS, Selleck W, Jayaram H, et al (2017) Characterization of the interplay between DNA repair and CRISPR/Cas9-induced DNA lesions at an endogenous locus. *Nat Commun* 8: 13905

Lee J, Lim K, Kim A, Mok YG, Chung E, Cho SI, Lee JM & Kim JS (2023). Prime editing with genuine Cas9 nickases minimizes unwanted indels. *Nat Commun* 14: 1786

Friedland AE, Baral R, Singhal P, Loveluck K, Shen S, Sanchez M, Marco E, Gotta GM, Maeder ML, Kennedy EM, et al (2015) Characterization of *Staphylococcus aureus* Cas9: a smaller Cas9 for all-in-one adeno-associated virus delivery and paired nickase applications. *Genome Biol* 16: 257

2. Line 127/128: In this passage it was not entirely clear to me why the dual nicking approach was chosen or switched to (from nuclease), in the first place. Reducing off-targets makes sense, but it reads here as if you detected substantial off-targets with the nuclease approach and therefore wanted to switch. Was that the case? If so, it would be great to describe in more detail what the off-targets looked like or how abundant they were with the nuclease approach.

The observation of the referee is accurate; in our initial study involving nuclease Cas9 (Zabaleta et al., 2018), no off-targets were identified. Nonetheless, it is important to highlight that the methodology employed to identify these off-targets was based on *in silico* predictions followed by targeted NGS analysis, a method that might not offer the highest accuracy. Furthermore, results regarding off-targets in mice should not be directly extrapolated to humans, and enhancing the safety of this system is crucial for translation to patients. In our recent work, we employed more sensitive techniques, such as CIRCLE-seq and CAST-Seq, to analyze the presence of off-targets. Using the same gRNAs and nuclease Cas9 as in the earlier study, we still detected no off-targets. Additionally, while off-targets were not identified in any of our treatments, we observed that the use of nickase Cas9 reduces the frequency of AAV integration compared to the use of nuclease Cas9, thereby increasing the safety of the proposed strategy for targeted gene disruption. We have added a short explanation in the introduction (Line 145-149) as well as in the discussion, in which we also highlighted the limitation of our study due to the specificity of our guides (Lines: 502-513).

Zabaleta N, Barberia M, Martin-Higuera C, Zapata-Linares N, Betancor I, Rodriguez S, Martinez-Turrillas R, Torella L, Vales A, Olagüe C, et al (2018b) CRISPR/Cas9-mediated glycolate oxidase disruption is an efficacious and safe treatment for primary hyperoxaluria type I. *Nat Commun* 9: 5454

3. Line 135: I think it would make sense to refer to SaCas9 nickase specifically here because there are some differences between e.g., SpCas9 and SaCas9 when it comes to mutations of the HNH domain. Therefore, I wouldn't necessarily mention N863A here, given that this was an initial HNH mutant for SpCas9, and H840A is by far the most commonly used mutant for this purpose nowadays. One might consider mentioning the alternative SaCas9 HNH nicking variant H557A though (Kleinstiver *et al*, 2015; Nishimasu *et al*, 2015).

The text has been modified, the SaCas9 HNH nicking variant H557A mentioned, and references introduced (Lines: 113-115).

Kleinstiver BP, Prew MS, Tsai SQ, Nguyen NT, Topkar VV, Zheng Z & Joung JK (2015) Broadening the targeting range of *Staphylococcus aureus* CRISPR-Cas9 by modifying PAM recognition. *Nat Biotechnol* 33: 1293–1298

Nishimasu H, Cong L, Yan WX, Ran FA, Zetsche B, Li Y, Kurabayashi A, Ishitani R, Zhang F & Nureki O (2015) Crystal Structure of *Staphylococcus aureus* Cas9. *Cell* 162: 1113–1126

4. Line 136: I think the authors swapped RuvC and HNH here.

D10 = RuvC, N580 = HNH.

Friedland et al., *Genome Biology* 2015: „We used site-directed mutagenesis to generate D10A and N580A mutants that would similarly disable the RuvC and HNH nuclease domains, respectively.”

We apologize for the mistake. The error has been corrected in the revised manuscript. (Lines: 113-115).

5. Line 194: What approach has higher vector dosing, nuclease or (current) dual-nick? Here, it sounds as if dual-nicking approach has lower dose but in line 233 it sounds like it's the other way around.

The referee is right the dose used for the dual-nick approach is ten and five times higher; we have modified the text accordingly. (Lines: 174-175).

6. Line 233: Could the authors please briefly explain why a 5x higher dose was needed for the dual-nicking approach compared to the WTSaCas9 nuclease?

In that specific experiment, 5×10^{13} vg/kg was the only tested dose. However, subsequent findings demonstrated that when using the all-in-one vector, a dose ten times lower exhibited a comparable editing efficacy to that of the SaCas9 combined with a single gRNA, and paired SaCas9 injected at a five times lower dose. Consequently, there is no genuine requirement for a dose higher than 5×10^{12} vg/kg.

7. Line 373: It would be helpful to explain why the translocation was detected in all experimental conditions but only classified as an OMT in the g2 treatment.

The reason why it is only classified as an OMT in the g2 treatment is that the pipeline looks for the highest read coverage from a particular putative off-target site, and then searches for a match to the target sequence within a window of +/- 100 bp. This was adapted from the T-CAST pipeline (Rhiel *et al* 2023) and makes the alignment more specific. However, in this situation, reads came from a very large region on chromosome 8, and the read coverage distribution in the different samples was not the same. Therefore, the window used for the alignment was not the same, and then by chance there was a region in the g2-treated samples that showed some similarity to the target sequence. Of note, even the predicted off-target site in chromosome 8 in the g2-treated samples had 4 mismatches and a deletion of 2 base pairs compared to the target site, and no matching PAM. For g1-treated samples, the target sequence is different and therefore in this case no match to the target sequence was identified.

We have changed the text to make this clearer (Lines: 352-354).

Rhiel M, Geiger K, Andrieux G, Rositzka J, Boerries M, Cathomen T & Cornu TI (2023) T-CAST: An optimized CAST-Seq pipeline for TALEN confirms superior safety and efficacy of obligate-heterodimeric scaffolds. *Front Genome Ed* 5: 1130736

8. Fig. 1a: I would give the PAMs a little "PAM" label. Also, cleavage is indicated at 2bp upstream of the PAM but probably is expected 3bp upstream for SaCas9, see Extended Data Fig. 2 of Ran et al, Nature 2015 (PMID 25830891).

We are sorry for the mistake. We have noticed the error and corrected it in the figure.

9. Fig. 1b: Is there a specific reason why the D10A-SaCas9 is thicker in the upper construct?

There is no need for a thicker structure. We have rectified it in the corresponding figure.

10. Fig. 2a/b: What do the labels left of the graph mean, e.g., 30:23D? Also, would it be possible to add allele frequencies here?

The labels left of the graph represent the location of the deletion with respect to the cleavage site and the size of the deletion. Thus, 30:23D means that the genetic modification is a deletion of 23 nucleotides at 30 nucleotides from the cleavage site of *Hao1*-gRNA1, which is identified by the black line in Figures 2A and 2B. The allele frequencies are reported in the respective Appendix Figures S3A and S4A.

Referee #3 (Remarks for Author):

In this study, Torella et al. explored the application of AAV-delivered paired SaCas9 nickase to disrupt the *Hao1* gene to treat primary hyperoxaluria type 1 (PH1). They showed that using an all-in-one AAV system to deliver SaCas9 D10A nickase and dual sgRNAs could effectively reduce the expression of glycolate oxidase (GO) enzyme, thus bringing about an obvious therapeutic effect in the mouse disease model. Importantly, this nickase strategy outperformed the conventional nuclease method in a significant reduction in AAV integration at the target site, which may be caused by different repair pathways determined by cutting ends. In general, this work provides important evidence showing the advantage of SaCas9 nickase approach, proposing an improved method for in vivo gene therapy. Despite the potential value, a part of current data is not solid enough to support the conclusions. More experiments should be performed before publication.

We thank the reviewer for the detailed assessment of our manuscript and the insightful suggestions and comments.

Major points:

(1) Line 195-198: As the editing efficiency of paired nickase method is a big concern, the authors did not strictly compare the efficiency of single, paired wild-type SaCas9 and paired SaCas9 nickase in one experiment. Since conditions vary between different experiments, comparing the current data with the previous ones is not reasonable.

The referee is right: our original main aim was to demonstrate that we could achieve the same editing frequency and GO depletion with SaCas9 nickase as with the WT SaCas9 while reducing the risk of off-target events. In the first experiment, we used a high dose of the vectors carrying the nickase Cas9 with each guide independently or in combination. We found that the editing frequency was similar to the one obtained with the WT SaCas9 and the two guides but, very importantly, no or very low editing was observed when using one guide and the nickase, which indirectly supports the notion that the probability of off-target events using the nickase is reduced because a single nick is repaired faithfully. A

DSB can only occur if two nicks on opposite strands and in close proximity are introduced. Then we performed a dose-response study and we found that we could significantly reduce the dose maintaining the editing efficacy. Although the experiments with paired WT SaCas9 nucleases and paired SaCas9 nickases were not performed side by side, the method used to analyze the editing efficacy (NGS and analysis done side-by-side) and the genomic modification was the same. We therefore believe that the data can be compared. Briefly, in all the experiments total genomic DNA was extracted from frozen liver sections using a NucleoSpin Tissue Extraction Kit, and nested PCR using the same primers was executed for NGS library preparation, as described in methods. In the different experiments, the 450 bp amplicons were purified as described above and sent to GENEWIZ for Amplicon-EZ analysis. The sequencing results were analyzed using the very same informatic pipeline.

Thus, we anticipate that repeating the experiment by injecting new animals with the same vectors will result in similar results in terms of gene editing frequencies and genomic modifications and will not provide additional insights. So, if the referee agrees we prefer not to perform additional *in vivo* studies. In fact, experiments are being designed to test, as the referee suggest in point number 3, if using other guides targeting the same locus or other loci and paired WT SaCas9 and nickase Cas9 raised similar results to the ones shown in the present manuscript. However, we first need to identify pairs of guides showing similar efficacy to the one observed in this work, which is not an easy task. Furthermore, production of the AAV vectors, *in vivo* experimentation, and genomic analysis will take more than 6 months.

I understand the authors' purpose to demonstrate the efficiency of paired SaCas9 nickase for the first step. As I mentioned above, the editing efficiency of paired nickase method is a big concern. Considering the high cost and long time for *in vivo* animal experiments, it can be accepted that the authors described the comparison between the current data and previous ones, although it is not exactly strict. Nevertheless, in my opinion, it is still necessary for the authors to show the comparison data of editing efficiency between single, paired wild-type SaCas9 and paired SaCas9 nickase in one assay. The authors can design a simple assay (e.g., *in vitro* assay) to detect the efficiency, especially at low doses to see whether the difference exists.

Based on the reviewer's recommendation, we conducted an *in vitro* study to compare the efficacy of single gRNA with paired wild-type SaCas9 and paired SaCas9 nickases. Due to the undetectable GO expression in murine hepatic cell lines (see Western Blot/mRNA analysis of Hepa 1-6 and AML12 cells in the figure below), we opted for HEK293T cells that were transfected with a plasmid encoding the mouse GO protein, which contains the gRNAs target sequences. Thus, cells were transfected with plasmids encoding single, paired wild-type SaCas9 and paired SaCas9 nickases, combined with the murine Hao1 plasmid. Heeding the reviewer's suggestion, we tested three different plasmid doses encoding the genome editors to assess the dose-response of both nuclease and nickase SaCas9. The results are presented in Appendix Figure S1 of the manuscript. WB analysis of GO expression under different conditions showed that both paired wild-type SaCas9 and paired SaCas9 nickases exhibited comparable efficiency across all doses. In contrast, the single gRNA1 displayed a marginally reduced efficiency. Importantly, our experimental setup did not allow for a thorough examination of dose effects as we did not notice a dose response with the plasmid concentrations assessed.

Hepa 1-6

AML12 and Hepa 1-6

Figure: Analysis of GO expression in murine hepatic cell lines. A. GO expression was analysed by western blot in Hepa 1-6 cells transfected with a plasmid carrying *Hao1* mRNA to express the murine GO protein or untransfected. No GO protein expression was detected in untransfected Hepa 1-6 cells. B. Western blot signal was quantified and normalized in relation to GAPDH levels. C. *Hao1* mRNA levels were analysed in transfected and untransfected Hepa 1-6 cells. D. *Hao1* mRNA levels were analysed in Hepa 1-6 cells and in an additional murine hepatic cell line BNL in which the *Hao1* levels were even lower.

(2) Although the authors carried out CIRCLE-seq and CAST-seq to detect potential off-target events and chromosomal translocations, the assays were performed in MEFs, which was irrelevant with the treatment strategy that the study proposed. Given that off-target activities are always determined by specific conditions, an *in vivo* detection assay is needed. Also, importantly, in order to conclude the low frequency of off-target events when using paired SaCas9 nickase, the authors must compare it with wild-type SaCas9 with one or two sgRNAs in parallel.

The use of MEFs, in which we achieve 100% transfection efficiency, and WT SaCas9 nucleases instead of SaCas9 nickases, represents in our view this kind of worst-case scenario to analyze off-targets. If, in this condition, in which double-strand breaks are introduced by the nuclease, we do not detect off-target events, the probability of off-target events will be even lower with the use of the nicking enzyme. Thus, the results of this worst-case scenario are shown in this manuscript.

The authors can add a description in the article to explain why they used MEFs as mentioned above.

We have included the reasons behind choosing these experimental conditions in the text (Lines 502-513).

On the other hand, experiments to determine off-targets by applying CAST-Seq analysis to liver samples obtained from paired SaCas9 nucleases and nickases targeting *Hao1* gene have been initiated. Preliminary data from these experiments are in line with the results observed in the *in vitro* setting and they will be part of an independent manuscript by Klermund et al., which contains additional data that demonstrates the sensitivity of the method.

The reason that these data will be shown in a separate manuscript focusing on using CAST-Seq on *in vivo* edited samples is that only a few methods exist to date that can detect off-target editing *in vivo*, and CAST-Seq has several distinct advantages over the previously published methods. CAST-Seq can be used to directly analyze genomic DNA (gDNA) extracted from cells that were edited *in vivo*, it requires only small amounts of input gDNA, and the protocol used for *in vivo* editing does not have to be changed for

a subsequent CAST-Seq analysis. Further, CAST-Seq can be performed at different time points following *in vivo* editing.

In fact, only three methods have so far been developed to detect off-target editing using *in vivo* edited samples. First, VIVO (verification of *in vivo* off-targets) employs CIRCLE-Seq followed by NGS-based targeted amplicon sequencing (Akcakaya et al., 2018). Using a promiscuous gRNA (called gP) deliberately designed to have many off-targets, VIVO demonstrated for the first time that *in vivo* CRISPR-Cas nuclease editing could lead to substantial off-target editing in mouse livers. However, VIVO method is indirect and identifies a large number of false positives. Second, DISCOVER-Seq (discovery of *in situ* Cas off-targets and verification by sequencing) is based on ChIP-Seq of Mre11, a DNA repair factor that binds to sites of DSBs (Wienert et al., 2019). While it is more specific than VIVO, the protocol requires precise timing to be able to detect transient Mre11 binding before DSB repair. DISCOVER-Seq+ improves the sensitivity by co-delivering an NHEJ inhibitor to increase residence time of Mre11 on genomic DNA (Zou et al., 2023). Third, GUIDE-tag is an adaptation of GUIDE-Seq for *in vivo* applications (Liang et al., 2022). While it shows high sensitivity, it relies on the co-delivery of a linear double-stranded DNA (dsDNA) to be captured into and thereby tag nuclease-mediated DSBs. Specifically, GUIDE-tag uses a modified Cas9 fused to monomeric streptavidin (Cas9-mSA) that binds to biotin-dsDNA. Both DISCOVER-Seq+ and GUIDE-tag therefore require changes in the editing protocols, limiting their implementation in preclinical research and rendering them impractical for patient biopsy analysis.

We used CAST-Seq to analyze gDNA from mouse liver cells that have been edited with the gRNA that was previously published and analysed with the different *in vivo* off-target analyses described above. The data from this analysis, along with a second gRNA and also the gRNAs published in Torella et al., will be published together to demonstrate the applicability of CAST-Seq for detecting off-targets in an *in vivo* setting.

Akcakaya P, et al. In vivo CRISPR editing with no detectable genome-wide off-target mutations. Nature 561, 416-419 (2018).

Wienert B, et al. Unbiased detection of CRISPR off-targets in vivo using DISCOVER-Seq. Science 364, 286-289 (2019).

Zou RS, et al. Improving the sensitivity of in vivo CRISPR off-target detection with DISCOVER-Seq. Nat Methods 20, 706-713 (2023).

Liang SQ, et al. Genome-wide detection of CRISPR editing in vivo using GUIDE-tag. Nat Commun 13, 437 (2022).

Figure for reviewers and associated text removed

As the authors wrote in the part of introduction, one of the main goals for the clinical use of gene editing approaches is to reduce the occurrence of off-target events. Accordingly, it is thought that the main purpose of using the paired SaCas9 nickase strategy by the authors is to decrease the off-target probability. And it is of importance for this study to address the safety concern by using SaCas9 nickase rather than SaCas9 nuclease. As the authors mentioned, by performing the assay in MEFs as a kind of worst-case scenario, they found no off-target events. Despite this desirable result, it becomes an issue that this data cannot demonstrate the advantage of using SaCas9 nickase to replace SaCas9 nuclease, because the deficiency of off-target events is attributed to the choice of superior sgRNAs but not the strategy per se. Also, as the below new data showed, no off-target mediated translocations were observed in any group. Thus, the potential advantage of reducing off-target events by the SaCas9 nickase strategy can not be proved as well, owing to the deficiency of negative data. Taken together, it is a double-edged sword problem that although the authors show the promising safety data, they cannot

demonstrate the superiority of the method. To address this, it may be available for the authors to choose other worse sgRNAs targeting *Hao1* gene to guarantee the observation of off-target effects and meanwhile compare the frequency between the wild-type SaCas9 and paired SaCas9 nickase methods.

On the other hand, the authors should add more description as they did in the rebuttal letter to help readers fully and clearly understand the meanings of the data and tone down the conclusions if they have no convincing data to prove the advantage of the SaCas9 nickase strategy in reducing off-target occurrence.

Given the high precision of the gRNAs tested, we could not underscore the enhanced safety of the paired nickases approach, although this has been demonstrated previously (Ran *et al*, 2013; Cho *et al*, 2014; Shen *et al*, 2014). Thus, to truly demonstrate the safety benefit of the nickase approach, it might be necessary to repeat the experiments using guides with demonstrated off-target activity. Importantly, the absence of detectable off-targets with the chosen gRNAs in mice might not necessarily extend to gRNAs targeting human *Hao1*, making it imperative to bolster the system of the editing safety before considering clinical applications in patients.

We have moderated our conclusions to clarify that while the paired nickase strategy is theoretically safer, our current experimental design did not provide conclusive evidence of this (Lines 551-563).

Ran FA, Hsu PD, Lin C-Y, Gootenberg JS, Konermann S, Trevino AE, Scott DA, Inoue A, Matoba S, Zhang Y, et al (2013) Double nicking by RNA-guided CRISPR Cas9 for enhanced genome editing specificity. *Cell* 154: 1380–1389.

Cho, SW, Kim, S, Kim, Y, Kweon, J, Kim, HS, Bae, S, et al. (2014). Analysis of off-target effects of CRISPR/Cas-derived RNA-guided endonucleases and nickases. *Genome Res* 24: 132-141.

Shen, B, Zhang, W, Zhang, J, Zhou, J, Wang, J, Chen, L, et al. (2014). Efficient genome modification by CRISPR-Cas9 nickase with minimal off-target effects. *Nat Methods* 11: 399-402.

(3) Since the significance of this study is to demonstrate paired SaCas9 nickase is a superior method for gene therapy, data of targeting more sites besides current two in the *Hao1* gene, showing comparable efficiency and reduced unintended AAV integration and off-target events, are indispensable. Moreover, considering this study only provided the evidence of effective treating PH1 by using paired SaCas9 nickase, it seems to be more proper to write "treating PH1" rather than "showing gene disruption" in the title.

We believe these studies are highly relevant as the referee indicates (Please also refer to the response provided for comment 1).

Our main aim was to demonstrate that the nickase system is as effective as Cas9 nucleases for the treatment of PH1 but safer due to a lower risk of off-target events. In fact, when using a single guide with the nickase, we barely observed any genetic modification. On the other hand, we observed that using the paired nickases approach resulted in a reduction of AAV integration.

We agree with the referee that in order to make general claims, it will be important to test the system at additional target sequences (Lines 502-513). As of today, we have no additional pair of guides with similar characteristics to the ones reported in this study (efficacy, distance, etc...) that will allow us to

perform this study in a reasonable timeframe. Furthermore, we have searched in the literature for additional studies using paired SaCas9 nickases delivered by AAV *in vivo* that we could use and did not find any reports. Thus, additional data would be needed to make a general claim. However, we refrain from making such general statements and limit our discussion to the specific case of treating PH1 by targeting this particular region in the *Hao1* gene. The discussion has been modified accordingly.

Following the referee's suggestion, we will modify the titer to "Efficient and Safe Therapeutic Use of Paired Cas9-Nickases for Primary Hyperoxaluria Type I"

I understand the difficulty for the authors to perform more experiments at other sites. The changes of the title and description can be accepted to address this comment.

We thank the reviewer for his understanding. We have changed the title of the manuscript to avoid confusion.

According to the responses below, I think these minor issues will be well addressed.

Minor points:

(1) In figure 1, the group setting is not consistent among figure 1C, 1D and 1E.

The setting has been corrected for consistency. We have added in figure 1 panel D the image of a mouse receiving the TBG-D10Ag2. However, NGS analysis was not performed in the animals receiving TBG-SaCas9 g1 since those data were reported in our previous publication (Zabaleta *et al.* 2018), thus we only included the western blot data from this group in panel 1C as control.

Zabaleta N, Barberia M, Martin-Higueras C, Zapata-Linares N, Betancor I, Rodriguez S, Martinez-Turrillas R, Torella L, Vales A, Olagüe C, *et al* (2018b) CRISPR/Cas9-mediated glycolate oxidase disruption is an efficacious and safe treatment for primary hyperoxaluria type I. *Nat Commun* 9: 5454

(2) Line 233: The authors mentioned "TBG-D10Ag1+g2-treated mice received five times more AAV vector genomes than those treated with TBG-SaCas9g1+g2", but detailed experimental conditions, such as dose and administration time, are deficient in the current sections of "results" and "materials and methods".

Additional information has been included in the revised version of the manuscript (Line 149-153)

(3) In figure 3B, the expression of GO looks higher in the high dose group than that in the median or low dose group, which is inconsistent with the result in figure 3C. Could the authors give an explanation?

The immunohistochemistry analysis revealed no GO expression in the group of animals treated with the high dose, while some hepatocytes in the groups treated with the median and low doses exhibited expression. These findings were further confirmed by western blot analysis. The immunohistochemistry images have been enhanced to present the data more accurately.

(4) Line 352: It is somewhat unclear for readers to know which off-target site is described by the authors in figure EV8. It will be helpful to add chromosome location for each off-target site.

We have labelled the off-target sites more clearly in Figure EV1 indicating the chromosomal location.

7th Nov 2023

Dear Dr. Gonzalez-Aseguinolaza,

Thank you for submitting your revised manuscript. We have now received the reports from the two referees who re-reviewed your manuscript, and as you will see below, they are supportive of publication. I will therefore be able to accept your manuscript once the following editorial points will be addressed:

1/ Manuscript text:

- Please remove the red text and only keep in track changes mode any new modification.
- We note that you currently have together with you, a total of 3 co-corresponding authors. Is that correct? Do you confirm equal contribution of these 3 people, able to take full responsibility for the paper and its content?
- We noted discrepancies in the following authors' name: Julen Torrens-Baile in the manuscript file vs. Julen Torrens in the submission system. Please clarify.
- Please provide up to 5 keywords.
- Materials and methods:
 - o Cells: please indicate the culture condition and whether the cells were tested for mycoplasma contamination.
 - o Statistics: please include a statement on sample size, blinding, randomization, exclusion/inclusion criteria.
- Data Availability section: Thank you for providing reviewers' token. Please note that the datasets must be public before acceptance of the manuscript.
- Acknowledgements: Please make sure that the information provided in the manuscript matches the information provided in the submission system (Fundación para la Investigación Médica Aplicada (FIMA) is currently missing from the submission system).
- Author contributions: CRediT has replaced the traditional author contributions section because it offers a systematic machine readable author contributions format that allows for more effective research assessment. Please remove the Authors Contributions from the manuscript and use the free text boxes beneath each contributing author's name in our system to add specific details on the author's contribution. More information is available in our guide to authors.
- Data Citation format: Please note that the data callout in the text for PRJNA488368 data citation does not include "Data ref:" as a prefix.

2/ Figures and Appendix:

- Please provide exact p values, not a range, in the figures or in their legends, including for ns - non-significant.
- The legend of Dataset EV1 needs to be removed from the manuscript and inserted in the Excel file as a separate sheet/tab.
- Appendix: please provide page numbers.

3/ Source Data:

Source Data have to be grouped and uploaded as one file per figure.

4/ Checklist:

- Please double-check that you need to fill in the part on Dual Use Research of Concern, as I don't think it applies to your manuscript.
- Please check and fill the right column wherever you filled the middle column in the checklist.

5/ Synopsis:

Please upload the image individually as a jpeg, TIFF or png file, 550 pixels wide x 200-400 pixels high, and make sure that the text remains legible.

6/ As part of the EMBO Publications transparent editorial process initiative (see our Editorial at <http://embomolmed.embopress.org/content/2/9/329>), EMBO Molecular Medicine will publish online a Review Process File (RPF) to accompany accepted manuscripts.

This file will be published in conjunction with your paper and will include the anonymous referee reports, your point-by-point response and all pertinent correspondence relating to the manuscript. Let us know whether you agree with the publication of the RPF and as here, if you want to remove or not any figures from it prior to publication. Similarly, do you want to remove the manuscript by Klermund et al at the end of the PbP?

I look forward to reading a new revised version of your manuscript as soon as possible.

Yours sincerely,

Lise Roth

***** Reviewer's comments *****

Referee #1 (Comments on Novelty/Model System for Author):

The manuscript has been improved significantly and all concerns raised were addressed.

Referee #1 (Remarks for Author):

The authors addressed all the concerns initially raised.

Referee #3 (Comments on Novelty/Model System for Author):

The application and clinical trial in human using the proposed method are anticipated in the future.

Referee #3 (Remarks for Author):

My comments have been well addressed by the authors.

The authors addressed the minor editorial issues.

15th Nov 2023

Dear Dr. Gonzalez-Aseguinolaza,

Thank you for providing your revised files. I am pleased to inform you that your manuscript is accepted for publication and is now being sent to our publisher to be included in the next available issue of EMBO Molecular Medicine!

Regarding the Review Process File, we note that you would like the removal of one figure and of the attached manuscript currently under review. We will prepare the RPF and send it to you for approval.

If you have any questions, please do not hesitate to contact the Editorial Office. Thank you for your contribution to EMBO Molecular Medicine and congratulations on your interesting work!

With kind regards,

Lise Roth
